# PPLLaVA: Varied Video Sequence Understanding With Prompt Guidance

## Abstract

The past year has witnessed the significant advancement of video-based large language models. However, the challenge of developing a unified model for both short and long video understanding remains unresolved. Most existing video LLMs cannot handle hour-long videos, while methods custom for long videos tend to be ineffective for shorter videos and images. In this paper, we identify the key issue as the redundant content in videos. To address this, we propose a novel pooling strategy that simultaneously achieves token compression and instruction-aware visual feature aggregation. Our model is termed Prompt-guided Pooling LLaVA, or PPLLaVA for short. Specifically, PPLLaVA consists of three core components: the CLIP-based visual-prompt alignment that extracts visual information relevant to the user's instructions, the prompt-guided pooling that compresses the visual sequence to arbitrary scales using convolution-style pooling, and the clip context extension designed for lengthy prompt common in visual dialogue. Moreover, our codebase also integrates the most advanced video Direct Preference Optimization (DPO) and visual interleave training. Extensive experiments have validated the performance of our model. With superior throughput, PPLLaVA achieves better results on image benchmarks as a video LLM, while achieving state-of-the-art performance across various video benchmarks, excelling in tasks ranging from caption generation to multiple-choice questions, and handling video lengths from seconds to hours. The codes are promised to be made public.

## 1 Introduction

Video Large Language Models (Video LLMs) have made significant advancements over the past year. Given the extensive resources and the scarcity of high-quality video-text data required for video pretraining, performing Image-to-Video transfer on powerful Image-domain Large Language Models has become a more practical approach for most Video LLMs. Building on the most advanced image LLMs (Liu et al., 2023a; 2024a; Dai et al., 2023), existing video LLMs typically address the modal differences between images and videos by video instruction data production (Maaz et al., 2023; Li et al., 2023b; Luo et al., 2023; Zhang et al., 2024b), temporal modeling (Liu et al., 2024c;d; Li et al., 2023c; Huang et al., 2023), or video token aggregation (Jin et al., 2023; Li et al., 2023d; Xu et al., 2024). Meanwhile, a wide range of video benchmarks and test tasks offer diverse perspectives and options for evaluating the capabilities of video LLMs, including video question answering (Maaz et al., 2023; Xu et al., 2016; Caba Heilbron et al., 2015; Wu et al., 2017), video dense captioning (Ren et al., 2024), multiple-choice questions (Li et al., 2023c; Fu et al., 2024), and long video assessment (Fu et al., 2024; Song et al., 2024; Zhang et al., 2024a).

For temporal modeling, an intuitive approach is to directly input tokens from each frame into the LLM, a method proven effective in several studies (Liu et al., 2024d;b; Li et al., 2024). However, while this method leverages the LLM's sequence modeling capabilities, it leads to an excessively long visual context. This not only increases computational resource consumption and processing time but also limits the model's ability to handle extended videos. To address this issue, several alternative approaches exist. A commonly adopted method is average pooling across the temporal dimension, frequently seen in early video LLMs (Li et al., 2023b; Maaz et al., 2023; Luo et al., 2023; Liu et al., 2024c). While this approach maintains a constant context length, it significantly diminishes the model's ability to capture temporal dynamics. Models designed specifically for long videos often incorporate unique structures, such as memory mechanisms (Ren et al., 2024; Zhang

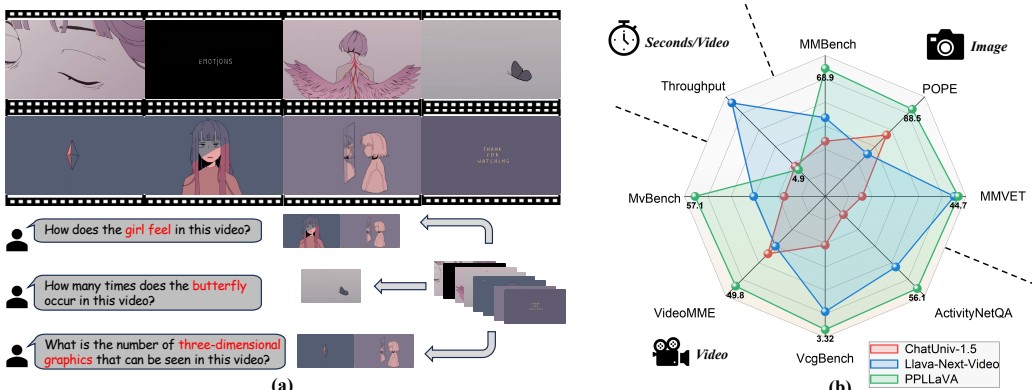

Figure 1: (a) An instance from VideoMME (Fu et al., 2024). The crucial information pertains to only a small portion of the video for different questions. (b) Performance comparison of PPLLaVA with recent strong Video LLM among video benchmarks, image benchmarks, and efficiency. All the models are based on Vicuna-7B.

et al., 2024a; Zhou et al., 2024). Although these designs enable the models to handle hour-long videos, they offer limited utility for short videos or images. Another approach is the use of conditional token pooling or aggregation (Li et al., 2023d; Xu et al., 2024; Jin et al., 2023). Unlike global average pooling, this method reduces the context length while preserving some spatiotemporal structure, enabling more effective spatiotemporal modeling.

However, pooling inevitably leads to performance loss compared to using the full set. So, how can we reduce the number of tokens while preserving the spatiotemporal modeling capabilities? We believe the solution lies in the inherent characteristics of the video. As proven by many previous works (Han et al., 2022; Liu et al., 2023b; Ma et al., 2022), videos contain significant redundancy, with key information often concentrated in just a few frames, which is particularly true for long videos. For video LLMs, this issue can be more pronounced. As shown in Fig. 1(a), the user's instruction may pertain to only a small portion of the video, with the rest being redundant for correctly answering the question. Therefore, if we can extract crucial video information while compressing tokens, we can maintain or even enhance performance. In this context, Image LLMs have offered valuable inspiration. The BLIP series (Li et al., 2023a; Dai et al., 2023; Xue et al., 2024) and the LLaVA series (Liu et al., 2023a; 2024a;b; Li et al., 2024) are the two most popular structures in multimodal LLM. BLIP uses a Q-Former for image-to-text mapping, while the LLaVA series employs simple linear projection or MLP. Recently, LLaVA-based models have demonstrated that simple mapping can achieve better results with less training (Liu et al., 2024a). However, despite requiring more computation resources and training stages, the Q-Former offers two key advantages: first, it significantly reduces visual tokens by converting them into fewer query tokens; second, through the interaction between text and visual tokens within the Q-Former, it enables more targeted extraction of video features relevant to the user's instructions (Dai et al., 2023). Hence, can we develop a pooling method that retains LLaVA's simple structure and powerful weights while reducing the number of tokens and enabling prompt-aware feature extraction?

To this end, we propose **P**rompt-guided **P**ooling **LLaVA** (PPLLaVA), a novel method that combines visual pooling with instruction-aware visual feature extraction. Specifically, PPLLaVA first identifies prompt-relevant visual representations through fine-grained vision-prompt alignment. Then, using the prompt-vision relevance as a 3D convolutional kernel, PPLLaVA can compress the visual tokens to any desired three-dimensional size based on the specified output size or stride. Finally, recognizing that CLIP pretraining provides a limited context length and that training video LLMs—particularly for multi-turn dialogues—requires long text contexts, PPLLaVA also employs asymmetric positional embedding extensions to expand the text encoding capacity. As a result, PPLLaVA effectively extracts relevant visual features from both long texts and short phrases while compressing video tokens. PPLLaVA achieves over an 80% compression rate, supports ultra-long video inputs, and simultaneously improves performance on short videos. In fact, PPLLaVA functions similarly to a Q-Former within LLaVA, but it offers several advantages over directly training a Q-Former: (1) PPLLaVA introduces far fewer additional parameters and computational overhead, amounting to less than one-tenth of a Q-Former. (2) While a Q-Former requires a three-stage

pretraining process—contrastive learning, alignment training, and instruction tuning—PPLLaVA can be utilized solely during instruction tuning, allowing for seamless transfer from image-domain LLMs. (3) PPLLaVA supports flexible output sizes for different modalities, whereas the number of queries in a Q-Former is fixed once set. As a result, different Q-Formers typically need to be trained separately for images and videos (Zhang et al., 2023; Li et al., 2023c).

Extensive experiments on the latest multimodal LLM benchmarks have validated the superiority of PPLLaVA: with superior throughput, PPLLaVA has achieved top results across a wide range of test sets, including MSRVTT (Xu et al., 2016), MSVD (Wu et al., 2017), ActivityNet (Caba Heilbron et al., 2015), VCG Bench (Maaz et al., 2023), MVBench (Li et al., 2023c), and Video-MME (Fu et al., 2024). These benchmarks encompass tasks such as video question answering, detailed video captioning, and video multiple-choice questions, with video lengths ranging from seconds to hours. Furthermore, our codebase has integrated cutting-edge video LLM techniques, including video Direct Preference Optimization and video-image-multiple image interleave training. As shown in Fig. 1(b), compared to recent top Video LLMs, PPLLaVA demonstrates clear advantages across both video and image benchmarks, while responding 7x faster than LLaVA-Next-Video-7B.

## 2 RELATED WORKS

**Image-domain LLMs.** Image-domain pretrained models have long served as the foundation for video understanding (Carreira & Zisserman, 2017; Luo et al., 2022; Liu et al., 2023c). This is partly due to the inherent similarities between image and video modalities and partly because image pre-training datasets offer a level of quality, quantity, and diversity that video datasets often lack. In the field of multimodal LLMs, the BLIP and LLaVA series have consistently served as the foundation for various video LLMs. The BLIP series is particularly notable for its Q-Former (Li et al., 2023a), which acts as an intermediary between the vision encoder and the LLM. The Q-Former not only enhances visual encoding but also compresses the number of visual tokens. Building on this foundation, InstructBLIP further developed the Q-Former's capability to extract instruction-aware visual features, making it a preferred choice for some video LLMs (Zhang et al., 2023; Li et al., 2023d; Liu et al., 2024d; Ren et al., 2024). LLaVA, a pioneer in visual instruction tuning (Liu et al., 2023a), accomplished the mapping from the visual encoder to the LLM using simple linear layers or MLPs. The LLaVA series has been continually updated (Liu et al., 2024a;b; Li et al., 2024), with later versions showing that this straightforward mapping approach can achieve superior results with less data. This simplicity and effectiveness inspired us to use LLaVA as the foundation for our model. Alongside this, we introduced the pooling module that retains LLaVA's efficient structure while also enabling the compression of visual tokens and the extraction of prompt-specific visual features.

**Video LLMs.** In the past year, Video LLMs have experienced rapid development since their inception. For video LLMs, updating video instruction data and benchmarks is essential. Video-ChatGPT (Maaz et al., 2023) was the first to introduce a high-quality video instruction training dataset and test set, establishing a benchmark for GPT-assisted evaluation. MVBench (Li et al., 2023c) provides a multiple-choice benchmark that assesses video performance across 20 different tasks. Video-MME (Fu et al., 2024) extends video duration significantly, reaching up to several hours, and serves as a comprehensive multiple-choice video QA benchmark. On the other hand, early Video LLMs (Li et al., 2023b; Zhang et al., 2023; Luo et al., 2023; Maaz et al., 2023; Liu et al., 2024c) typically used average pooling to process video sequences with Image LLMs while employing modality perceivers to model temporal sequences. However, this approach significantly limited the model's ability to fully understand video sequences. Alternatively, some models (Liu et al., 2024d;b; Xu et al., 2024) rely on the LLM itself to model video sequences, achieving good video understanding results. Nonetheless, this method is limited to handling a small number of frames and does not support the comprehension of long videos.

Understanding long videos is also a hot topic in video LLMs. MovieChat (Song et al., 2024) and Flash-VStream (Zhang et al., 2024a) use memory structures to process streaming videos, while Chat-UniVi (Jin et al., 2023) adopts a clustering approach for token aggregation. LLaMA-VID (Li et al., 2023d) compresses each video frame into two tokens, capturing both local and global information. Most similar to our work, PLLaVA (Xu et al., 2024) employs a non-parametric AdaptiveAvgPool3d function to compress visual tokens. In contrast, our method supports not only token compression but also the extraction of visual features pertinent to user prompts. Furthermore, our convolution-style pooling method enables flexible output sizes. Notably, compared to the aforementioned methods,

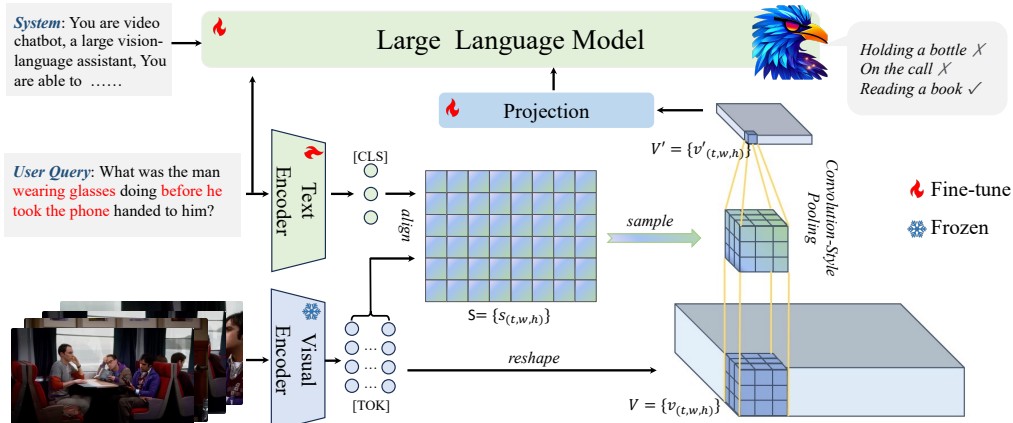

Figure 2: The overview of PPLLaVA for compressing the video based on user prompts and generating responses on the input video and instructions.

our approach has achieved state-of-the-art results on both long and short video benchmarks, whereas the other methods may exhibit slightly lower performance on videos of certain lengths.

The diversification of data modalities and formats has also become a prominent direction in research. Beyond the classic image-video instruction tuning, CAT (Ye et al., 2024) introduced mixed training with video and audio, while VideoMME (Fu et al., 2024) emphasized the importance of subtitles. VILA (Lin et al., 2024) and LLaVA-Interleave underscored the value of interleaved training. Besides instruction tuning, Reinforcement Learning from Human Feedback (RLHF) has also been proven to be particularly effective for video LMM. Specifically, VLM-RLAIF (Ahn et al., 2024) and LLaVA-Hound (Zhang et al., 2024b) demonstrated the effectiveness of Proximal Policy Optimization (PPO) and Direct Preference Optimization (DPO), respectively. We have also integrated these cutting-edge techniques into our codebase and demonstrated that they can operate in parallel with PPLLaVA.

## 3 METHODOLOGY

### 3.1 MOTIVATION AND ANALYSIS

In the previous section, we discussed that the videos are redundant in both length and content. Vista-LLaMA (Ma et al., 2024) demonstrated that the extensive number of tokens in long videos makes it difficult for LLMs to capture video content. In this section, we further examine whether redundant video content impacts the performance of video LLMs and whether extracting key video content can enhance performance. Inspired by EgoSchema (Mangalam et al., 2024), we adopt the certificate length to measure the redundancy. The certificate length of a video-QA pair is determined by the shortest video sub-clip that can answer the question. Instead of using manual annotation, we employed an automated method to determine the certificate. Specifically, frames are sampled at 2 fps, and then the similarity between each frame and the question-answer text is calculated using CLIP-L-336 (Radford et al., 2021). If the similarity exceeds 0.5, the frame is considered relevant to the text. Finally, the proportion of relevant frames to the entire video is calculated as the certificate.

Based on the Video-MME dataset, we selected the 100 video-QA pairs with the shortest certificate lengths termed Video-MME-redund. We then evaluated the performance of various models on both the full Video-MME dataset and these selected samples. Additionally, for these 100 samples, we manually selected the frames most relevant to the questions, alongside the default frame sampling method. This approach was used to test whether extracting key information enhances video understanding. As shown in Table 1, all models experienced a decline

Table 1: The study on the impact of video redundancy, we used the Vicuna-7B version for all models. "Average" and "Manual" refer to the default average frame sampling and manual frame selection, respectively.

| Model | Frames | Tokens | Video-MME-full average | Video-MME-redund | |
|---|---|---|---|---|---|
| | | | | average | manual |
| InstructBLIP | 32 | 1024 | 39.2 | 36.1 (-3.1) | 39.5 (+0.3) |
| LLaVA-Next | 32 | 4608 | 41.1 | 36.9 (-4.2) | 42.0 (+0.9) |
| LLaVA-Next-Video | 8 | 1152 | 42.9 | 39.0 (-3.9) | 43.5 (+0.6) |
| LLaVA-Next-Video | 32 | 4608 | 45.0 | 41.5 (-3.5) | 46.1 (+1.1) |
| PPLLaVA (ours) | 32 | 1024 | 49.8 | 47.6 (-2.2) | 50.5 (+0.7) |

in performance on high-redundancy videos. As an earlier model, InstructBLIP performed as expected, not matching the overall performance of the more advanced LLaVA-Next. However, on high-redundancy videos, InstructBLIP, which has instruction-aware video feature extraction capabilities, declined slower than LLaVA-Next. Furthermore, when manually selected frames were used, all models showed significant performance improvements, highlighting the importance of extracting key video information for enhancing video understanding. Additionally, we clearly observed the importance of including more frames for long videos, such as those in the Video-MME dataset. These findings motivated us to explore token compression to accommodate more video frames while effectively extracting key information.

## 3.2 PPLLAVA

As shown in Fig. 2, PPLLaVA, like most video LLMs, includes a vision encoder, a mapping layer, and a LLM. It also features an additional text encoder paired with the visual encoder. Given a $T$-frame video, we first pass it through the CLIP-ViT visual encoder, obtaining the visual feature $V \in \mathbb{R}^{T \times W \times H \times D}$. This feature is then fed into the Prompt-guided Pooling module, where it is compressed by over 90%, resulting in $V' \in \mathbb{R}^{T' \times W' \times H' \times D}$. $V'$ is fed into the MLP mapping layer as the final visual input. Importantly, $V'$ not only contains significantly fewer tokens but also condenses information more relevant to the user's instructions. This ensures improved performance while efficiently processing the video input. Next, we will detail how $V'$ is obtained.

**Fine-grained Vision-Prompt Alignment.** To extract video features relevant to the prompt, we first utilize the original CLIP dual encoders to identify which video features are related to the text. Specifically, we input the user's question into the CLIP text encoder to obtain the text feature $c \in \mathbb{R}^D$. Following the CLIP training pipeline, we only use the CLS token of the text. The attention score of the $(t^{th}, w^{th}, h^{th})$ video token relative to the text feature is then calculated as:

$$s_{(t,w,h)} = \frac{\exp(\tau c \cdot f_{clipv}(v_{(t,w,h)}))}{\sum_{t=1}^{T} \sum_{w=1}^{W} \sum_{h=1}^{H} \exp(\tau c \cdot f_{clipv}(v_{(t,w,h)}))}, \quad (1)$$

where $v_{(t,w,h)}$ represents the token at the $(t, w, h)$ position in $V$, $\tau$ is the CLIP temperature scale, and $f_{clipv}$ is the CLIP visual projection, which is typically not used in multimodal LLMs. Note that $v_{(t,w,h)}$ typically refers to the patch token from the penultimate layer of CLIP, rather than the CLS token from the final layer used during CLIP training. However, since the spatial representations in CLIP's final layers are similar, applying $f_{clipv}$ still allows the patch tokens to be mapped into the interaction space with the text.

**Prompt-Guided Pooling.** In the previous section, we obtained token-level weights corresponding to the user's prompt, which we use as guidance for pooling the video. Unlike traditional tasks that require only a D-dimensional feature for contrastive learning (Ma et al., 2022; Wang et al., 2022), our approach aims to preserve a certain 3-dimensional structure to enable the LLM to perform temporal modeling. To achieve this, we perform pooling with $S = \{s_{(t,w,h)}\}$ in a manner similar to 3D convolution. Specifically, we define the spatiotemporal 3D convolution kernel and stride as $(k_t, k_w, k_h)$ and $(d_t, d_w, d_h)$, respectively. The output dimension of $V'$ can then be expressed as:

$$T' = (\frac{T - k_t}{d_t}) + 1, \quad W' = (\frac{W - k_w}{d_w}) + 1, \quad H' = (\frac{H - k_h}{d_h}) + 1. \quad (2)$$

Unlike conventional convolution kernels, our kernel parameters are derived from $S$. Moreover, the parameters of the kernel are dynamic; as the kernel slides over different positions in $V$, its parameters are taken from the corresponding positions in $S$. Finally, the feature at position $(t, w, h)$ in the output $V'$ is calculated as:

$$v'_{(t,w,h)} = \sum_{i=0}^{k_t-1} \sum_{j=0}^{k_w-1} \sum_{k=0}^{k_h-1} v_{(t*d_t+i, w*d_w+j, h*d_h+k)} s_{(t*d_t+i, w*d_w+j, h*d_h+k)}. \quad (3)$$

By flexibly adjusting the stride and kernel size, we can control the output dimensions. This approach allows us to better accommodate videos of varying lengths and facilitates joint training with images, compared to fixed-output methods.

**CLIP Context Extension.** In our method, CLIP-text is the only additional parameter used. Despite having significantly fewer parameters than Qformer, it achieves better performance. However,

CLIP-text has a major limitation: its context length is too short (default is 77). While this length is sufficient for objects or simple descriptions, it is inadequate for long prompts or multi-turn dialogues in multimodal LLMs. To address this performance bottleneck, we propose extending the context length of CLIP-text using asymmetric positional embedding extensions. In most cases, extending the positional embedding involves randomly initializing new embeddings at the end. A more theoretically sound approach is to perform linear interpolation on the original positional embedding at a rate of $r$. Assuming the original and target positional embeddings are $P$ and $P'$, respectively, the $i^{th}$ position of $P'$ can be represented as:

$$P'_i = P_{\lfloor j \rfloor} + (j - \lfloor j \rfloor) \cdot (P_{\lfloor j \rfloor+1} - P_{\lfloor j \rfloor}), \;\; j = i \cdot r, \tag{4}$$

where $\lfloor j \rfloor$ means taking the floor of $j$. However, we found linear interpolation yielded inferior results to randomly initializing embeddings at the end. We believe this is because CLIP's positional embeddings are well-trained, and globally averaged interpolation disrupts the well-pre-trained information. Given that short sentences dominate CLIP's training data, the earlier parts of positional embeddings are more thoroughly trained. Hence, we adopted asymmetric interpolation, applying different interpolation rates at different positions. In the early part of the new positional embedding, we use a large $r$ value to shorten the interpolation distance, while in the later part, we use a smaller $r$ value to extend the interpolation distance. This asymmetric approach allows us to effectively extend the context length of CLIP-text while preserving as much of the pre-trained information as possible.

### 3.3 Training

**Interleave Instruction Tuning.** PPLLaVA enables plug-and-play transfer of image-domain LLMs to the video domain. As a result, initialized from well-pretrained image LLM, we can bypass expensive contrastive or alignment pretraining and proceed directly to instruction tuning. In this stage, we fully fine-tune the LLM, the projection MLP, and the CLIP text encoder. Our instruction datasets include multi-turn and single-turn conversations presented in a conversational format, along with various forms of visual input such as images, videos, and multiple images. For different types of data, we employed an interleaving training approach. Rather than using batches composed of a single data type, we mixed various data types within the same batch. Numerous studies (Li et al., 2024; Laurençon et al., 2024; Xue et al., 2024) have demonstrated that this method is the most natural approach for handling multimodal data. Additionally, this training method enables the model to simultaneously process both long videos with many frames and single-frame images, greatly enhancing its adaptability to visual sequences of varying lengths.

**Direct Preference Optimization. (DPO)** Video, especially long video-based dialogue, is more prone to hallucinations compared to images. As a result, Reinforcement Learning from Human Feedback (RLHF) (Zhang et al., 2024b; Ahn et al., 2024) has proven particularly effective for video. Therefore, we also implemented this method based on our model. Following LLaVA-Hound (Zhang et al., 2024b). We used detailed video captions as proxies for video content and performed DPO with feedback from the language model serving as a reward. In this stage, all parameters except the LLM were frozen, and only video data was used. This additional phase significantly reduced the occurrence of hallucinations during video-based dialogue.

## 4 Experiments

In this section, we have performed comprehensive experimental evaluations of PPLLaVA, covering crucial settings, comparisons, and ablations, while more ablation studies, visualizations, and limitations analysis can be found the appendix.

### 4.1 Experiment Setup

**Implementation Details.** PPLLaVA is built upon the advanced image-domain LLaVA-Next models (Liu et al., 2024b). To ensure a fair comparison with most models, we chose the Vicuna-7B version. For image and multiple-image inputs, the pooling kernel and strides are set to $(1, 3, 3)$. For video inputs, we uniformly sample 32 frames and set the pooling kernel and strides to $(2, 3, 3)$, compressing the video tokens by over 15 times. During training, both questions and answers are fed into the CLIP text encoder to better capture prompt-vision relevance. For CLIP context extension, when $i < 20$, $r$ is set to 1, and when $i \geq 20$, $r$ is set to 0.25. We train for one epoch using a learning rate

Table 2: The results of open-ended QA with GPT-based evaluation, including MSVD-QA, MSRTT-QA, ActivityNet-QA(ANet), and VCG Bench. All the models are based on the Vicuna-7B. † means using DPO or PPO.

| Method | MSVD | | MSRVTT | | ANet | | VCG Bench | | | | | |
|---|---|---|---|---|---|---|---|---|---|---|---|---|
| | Acc. | Sco. | Acc. | Sco. | Acc. | Sco. | CI | DO | CU | TU | CO | Avg. |
| VideoChat (Li et al., 2023b) | 56.3 | 2.8 | 45.0 | 2.5 | 26.5 | 2.2 | 2.23 | 2.50 | 2.53 | 1.94 | 2.24 | 2.29 |
| Video-ChatGPT (Maaz et al., 2023) | 64.9 | 3.3 | 49.3 | 2.8 | 35.2 | 2.7 | 2.50 | 2.57 | 2.69 | 2.16 | 2.20 | 2.42 |
| BT-Adapter (Liu et al., 2024c) | 67.5 | 3.7 | 57.0 | 3.2 | 45.7 | 3.2 | 2.68 | 2.69 | 3.27 | 2.34 | 2.46 | 2.69 |
| Video-LLaVA (Lin et al., 2023) | 70.7 | 3.9 | 59.2 | 3.5 | 45.3 | 3.3 | - | - | - | - | - | - |
| MovieChat (Song et al., 2024) | 75.2 | 3.8 | 52.7 | 2.6 | 45.7 | 3.4 | 2.76 | 2.93 | 3.01 | 2.24 | 2.42 | 2.67 |
| Chat-UniVi (Jin et al., 2023) | 65.0 | 3.6 | 54.6 | 3.1 | 45.8 | 3.2 | 2.89 | 2.91 | 3.46 | 2.89 | 2.81 | 2.99 |
| VideoChat2 (Li et al., 2023c) | 70.0 | 3.9 | 54.1 | 3.3 | 49.1 | 3.3 | 3.02 | 2.88 | 3.51 | 2.66 | 2.81 | 2.98 |
| Vista-LLaMA (Ma et al., 2024) | 65.3 | 3.6 | 60.5 | 3.3 | 48.3 | 3.3 | 2.44 | 2.64 | 3.18 | 2.26 | 2.31 | 2.57 |
| LLaMA-VID (Li et al., 2023d) | 69.7 | 3.7 | 57.7 | 3.2 | 47.4 | 3.3 | 2.96 | 3.00 | 3.53 | 2.46 | 2.51 | 2.89 |
| ST-LLM (Liu et al., 2024d) | 74.6 | 3.9 | 63.2 | 3.4 | 50.9 | 3.3 | 3.23 | 3.05 | 3.74 | 2.93 | 2.81 | 3.15 |
| PLLaVA (Xu et al., 2024) | 76.6 | 4.1 | 62.0 | 3.5 | 56.3 | 3.5 | 3.21 | 2.86 | 3.62 | 2.33 | 2.93 | 2.99 |
| CAT (Ye et al., 2024) | - | - | 62.1 | 3.5 | 50.2 | 3.5 | 3.08 | 2.95 | 3.49 | 2.81 | 2.89 | 3.07 |
| VLM-RLAIF † (Ahn et al., 2024) | 76.4 | 4.0 | 63.0 | 3.4 | 57.3 | 3.5 | 3.85 | 3.45 | 3.84 | 3.63 | 2.8 | 3.49 |
| LLaVA-Next-Video (Liu et al., 2024b) | - | - | - | - | 53.5 | 3.2 | 3.39 | 3.29 | 3.92 | 2.60 | 3.12 | 3.26 |
| LLaVA-Next-Video † | - | - | - | - | 60.2 | 3.5 | 3.64 | 3.45 | 4.17 | 2.95 | **4.08** | 3.66 |
| PPLLaVA | 75.8 | 3.9 | 61.9 | 3.3 | 56.1 | 3.4 | 3.32 | 3.20 | 3.88 | 3.00 | 3.20 | 3.32 |
| PPLLaVA † | **77.1** | 4.0 | **64.3** | 3.5 | **60.7** | 3.6 | **3.85** | **3.56** | **4.21** | **3.21** | 3.81 | **3.73** |

Table 3: Performance on Video-MME with short, medium, and long durations, under the settings of "without subtitles" and "with subtitles". * means using multi-images during training.

| Models | LLM Params | Short (%) | | Medium (%) | | Long (%) | | Overall (%) | |
|---|---|---|---|---|---|---|---|---|---|
| | | w/o subs | w/ subs | w/o subs | w/ subs | w/o subs | w/ subs | w/o subs | w/ subs |
| Qwen-VL-Chat (Bai et al., 2023) | 7B | 46.9 | 47.3 | 38.7 | 40.4 | 37.8 | 37.9 | 41.4 | 41.9 |
| Qwen-VL-Max (Bai et al., 2023) | - | 55.8 | 57.6 | 49.2 | 48.9 | 48.9 | 47.0 | 51.3 | 51.2 |
| InternVL-V1.5 (Chen et al., 2024) | 20B | 60.2 | 61.7 | 46.4 | 49.1 | 45.6 | 46.6 | 50.7 | 52.4 |
| Video-LLaVA | 7B | 45.3 | 46.1 | 38.0 | 40.7 | 36.2 | 38.1 | 39.9 | 41.6 |
| ST-LLM | 7B | 45.7 | 48.4 | 36.8 | 41.4 | 31.3 | 36.9 | 37.9 | 42.3 |
| VideoChat2-Mistral | 7B | 48.3 | 52.8 | 37.0 | 39.4 | 33.2 | 39.2 | 39.5 | 43.8 |
| Chat-UniVi-V1.5 | 7B | 45.7 | 51.2 | 40.3 | 44.6 | 35.8 | 41.8 | 40.6 | 45.9 |
| LLaVA-NeXT-Video | 7B | 45.9 | 49.8 | 40.3 | 44.3 | 36.6 | 41.0 | 40.9 | 45.0 |
| LLaVA-NeXT-Video | 34B | 61.7 | 65.1 | 50.1 | 52.2 | 44.3 | 47.2 | 52.0 | 54.9 |
| PPLLaVA | 7B | 56.1 | 59.7 | 43.9 | 48.6 | 38.4 | 44.0 | 46.1 | 50.0 |
| PPLLaVA* | 7B | **58.7** | **62.8** | **45.6** | **50.4** | **42.2** | **47.4** | **48.8** | **53.6** |

of $2e-5$ and a batch size of 256. We provid both GPU and NPU versions, and the full training takes 24 hours on 16 A100 GPUs or 32 910B NPUs.

**Data Details.** The instruction tuning data includes diverse modalities and sources. We randomly sampled 300k image data from the LLAVA-1.5 training set (Liu et al., 2024a) and used 594k multiple-image data from LLAVA-Interleave (Liu et al., 2024b). The video data includes Kinetics (Kay et al., 2017), SthSth-V2 (Goyal et al., 2017), Next-QA (Xiao et al., 2021), CLEVRER (Yi et al., 2019), and LLAVA-Interleave-300k, resulting in a total of 1.36M multimodal training samples. Notably, to ensure fairness in the comparison experiments, we excluded multi-image data and used only 760k image-video data, comparable to the training volume of most video LLMs.

We evaluate our model on six video LLM benchmarks, categorized into two types based on the evaluation method: GPT-based evaluation and multiple-choice questions. The GPT evaluation mainly involves open-ended QA, including the Video-based Generative Performance Benchmark (VCG Bench) (Maaz et al., 2023), MSVD-QA (Wu et al., 2017), MSRTT-QA (Xu et al., 2016), and ActivityQA (Caba Heilbron et al., 2015). Consistent with most models, we used the GPT-3.5-turbo-0613 version for testing. The multiple-choice question benchmarks include MVBench (Li et al., 2023c) and Video-MME (Fu et al., 2024). This evaluation method is more objective by eliminating the potential disturbances of GPT. For medium-to-long videos in Video-MME, we sampled 64 frames instead of the 32 frames used in other datasets. Our test corpus encompasses videos of various genres and lengths, offering a comprehensive evaluation of PPLLaVA's performance.

### 4.2 QUANTITATIVE RESULT

**GPT-Based Evaluation.** Table 2 presents the quantitative results for open-ended question-answering, showing that PPLLaVA achieves top performance across all datasets. It also demonstrates a significant performance gap compared to models other than LLaVA-Next-Video, demon-

Table 4: Results on MVBench. Models without additional annotation are 7B by default. * means using multi-images during training.

| Method | AS | AP | AA | FA | UA | OE | OI | OS | MD | AL | ST | AC | MC | MA | SC | FP | CO | EN | ER | CI | Avg. |
|---|---|---|---|---|---|---|---|---|---|---|---|---|---|---|---|---|---|---|---|---|---|
| Video-LLaMA | 27.5 | 25.5 | 51.0 | 29.0 | 39.0 | 48.0 | 40.5 | 38.0 | 22.5 | 22.5 | 43.0 | 34.0 | 22.5 | 32.5 | 45.5 | 32.5 | 40.0 | 30.0 | 21.0 | 37.0 | 34.1 |
| LLaMA-Adapter | 23.0 | 28.0 | 51.0 | 30.0 | 33.0 | 53.5 | 32.5 | 33.5 | 25.5 | 21.5 | 30.5 | 29.0 | 22.5 | 41.5 | 39.5 | 25.0 | 31.5 | 22.5 | 28.0 | 32.0 | 31.7 |
| Video-ChatGPT | 23.5 | 26.0 | 62.0 | 22.5 | 26.5 | 54.0 | 28.0 | 40.0 | 23.0 | 20.0 | 31.0 | 30.5 | 25.5 | 39.5 | 48.5 | 29.0 | 33.0 | 29.5 | 26.0 | 35.5 | 32.7 |
| VideoChat | 33.5 | 26.5 | 56.0 | 33.5 | 40.5 | 53.0 | 40.5 | 30.0 | 25.5 | 27.0 | 48.5 | 35.0 | 20.5 | 42.5 | 46.0 | 26.5 | 41.0 | 23.5 | 23.5 | 36.0 | 35.5 |
| VideoChat2 | 66.0 | 47.5 | **83.5** | 49.5 | 60.0 | 58.0 | 71.5 | **42.5** | 23.0 | 23.0 | **88.5** | 39.0 | 42.0 | 58.5 | 44.0 | 49.0 | 36.5 | 35.0 | 40.5 | **65.5** | 51.1 |
| ST-LLM | 66.0 | 53.5 | 84.0 | 44.0 | 58.5 | 80.5 | 73.5 | 38.5 | **42.5** | 31.0 | 86.5 | 36.5 | 56.5 | 78.5 | 43.0 | 44.5 | 46.5 | 34.5 | 41.5 | 58.5 | 54.9 |
| PLLaVA-7B | 58.0 | 49.0 | 55.5 | 41.0 | 61.0 | 56.0 | 61.0 | 36.0 | 23.5 | 26.0 | 82.0 | 39.5 | 42.0 | 52.0 | 45.0 | 42.0 | 53.5 | 30.5 | 48.0 | 31.0 | 46.6 |
| PLLaVA-13B | 66.0 | 53.0 | 65.5 | 45.0 | 65.0 | 58.0 | 64.5 | 35.5 | 23.5 | 30.0 | 85.0 | **39.5** | 45.5 | 57.0 | 47.5 | **49.5** | 49.0 | 33.0 | **53.0** | 37.0 | 50.1 |
| PLLaVA-34B | 67.5 | 53.0 | 82.0 | 47.0 | 79.0 | 68.5 | 67.5 | 36.5 | 37.5 | 49.5 | 91.0 | 40.5 | 43.0 | 70.0 | 51.5 | 50.0 | 66.5 | 39.5 | 63.5 | 59.0 | 58.1 |
| GPT-4V | 55.5 | **63.5** | 72.0 | 46.5 | **73.5** | 18.5 | 59.0 | 29.5 | 12.0 | **40.5** | 83.5 | 39.0 | 12.0 | 22.5 | 45.0 | 47.5 | 52.0 | 31.0 | 59.0 | 11.0 | 43.5 |
| PPLLaVA-7B | 69.0 | 54.4 | 69.5 | **50.5** | 69.0 | 87.0 | 67.0 | 38.0 | 35.0 | 33.0 | 69.5 | 37.5 | 63.5 | 91.0 | 47.5 | 47.5 | 51.5 | 27.0 | 47.5 | 57.5 | 57.1 |
| PPLLaVA-7B* | **73.5** | 61.0 | **83.5** | 45.5 | 68.0 | **87.5** | **75.5** | 33.0 | 37.5 | 40.0 | 83.0 | 37.0 | 67 | **96.5** | **50.5** | 43.5 | **59.0** | 35.5 | 44.5 | 63.0 | **59.2** |

Table 5: The ablation study on model components. TP means throughput (seconds/video).

| Model | Context Length | VCG Bench | | | | | | | Video-MME (w/ subs) | | | | |
|---|---|---|---|---|---|---|---|---|---|---|---|---|---|
| | | CI | DO | CU | TU | CO | Avg | TP | Short | Medium | Long | Overall | TP |
| LLaVA-Next (Average Pooing) | 576 | 3.05 | 3.07 | 3.71 | 2.62 | 3.01 | 3.09 | 2.9 | 53.1 | 41.3 | 36.0 | 43.4 | 3.1 |
| LLaVA-Next (w/o Pooing) | 4608 | 3.23 | 3.08 | 3.82 | 2.75 | 3.11 | 3.20 | 15.0 | 58.4 | 45.1 | 38.8 | 47.4 | 15.2 |
| +Prompt-guided Pooling | 1024 | 3.21 | 3.15 | 3.80 | 2.88 | 3.02 | 3.21 | 4.6 | 59.0 | 45.6 | 42.2 | 48.9 | 5.3 |
| +CLIP Context Extension | 1024 | 3.32 | 3.20 | 3.88 | 3.00 | 3.20 | 3.32 | 4.6 | 59.7 | 48.6 | 44.0 | 50.0 | 5.3 |

strating the strong text generation capability of our model. Despite using lower-quality data (as LLaVA 1.6 data is not publicly available), PPLLaVA outperforms LLaVA-Next-Video. More importantly, PPLLaVA uses significantly fewer visual contexts (1024 vs. 4096), resulting in higher throughput. After applying DPO, PPLLaVA also shows consistent improvements and outperforms other models that use DPO or PPO, further proving the adaptability of the PPLLaVA architecture across different training stages.

**Video-MME.** Although Video-MME is a new benchmark, it offers high quality and data diversity. Its inclusion of hour-long videos makes it particularly effective for evaluating models' long video understanding capabilities. As shown in Table 3, PPLLaVA achieves the best results on Video-MME, with a notably significant advantage on videos of different lengths compared to other models. The 7B model's long video comprehension already surpasses the 34B LLaVA-Next-Video, as PPLLaVA efficiently compresses video tokens, enabling support for a much higher number of frames than LLaVA-Next-Video, thereby enhancing long video understanding capabilities.

**MVBench.** MVBench is a multiple-choice benchmark offering a comprehensive set of evaluation tasks, including Action Sequence (AS), Action Prediction (AP), Action Antonym (AA), Fine-grained Action (FA), Unexpected Action (UA), Object Existence (OE), Object Interaction (OI), Object Shuffle (OS), Moving Direction (MD), Action Localization (AL), Scene Transition (ST), Action Count (AC), Moving Count (MC), Moving Attribute (MA), State Change (SC), Fine-grained Pose (FP), Character Order (CO), Egocentric Navigation (EN), Episodic Reasoning (ER), Counterfactual Inference (CI), and the average across all 20 metrics (Avg). PPLLaVA achieves the best average results among models, demonstrating a clear advantage and strong adaptability in video understanding across diverse scenarios, especially for moving and action tasks.

## 4.3 ABLATIONS AND ANALYSIS

**Model Components.** The core of PPLLaVA is its prompt-guided token compression, which enhances both video understanding efficiency and performance. To assess the impact of this feature, we conducted ablation experiments on the overall model components. As shown in Table 5, while the LLaVA-Next Baseline's direct averaging method is the most efficient, its performance is subpar. Directly feeding all tokens into the LLM yields reasonable results but suffers from extremely low throughput. Our Pooling module substantially improves both efficiency and performance. Extending the CLIP context further enhances results, particularly in long video understanding. The simultaneous improvement in efficiency and effectiveness underscores the superiority of our model.

**Pooling Size.** PPLLaVA can flexibly implement pooling at any scale. However, as the pooling kernel and stride increase, while efficiency improves, there will inevitably be performance degradation. Therefore, it's crucial to find a pooling size that balances both efficiency and performance. As illustrated in Fig. 3, we first explore the impact of pooling in the spatial dimension. It is evident that when the pooling kernel and stride are small, increasing them significantly improves efficiency,

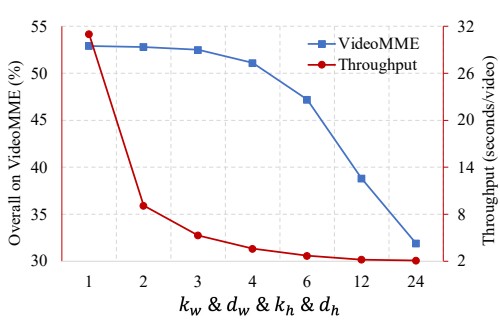 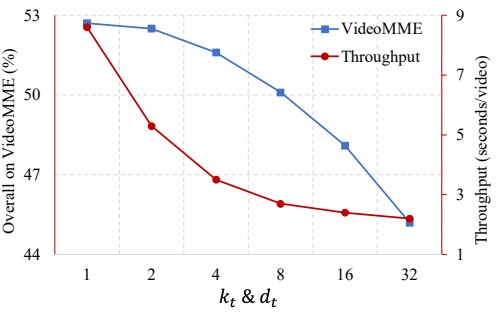

Figure 3: Spatial pooling effects. We set $T = 16$ and $k_t = d_t = 1$, varying the spatial kernel size and stride.

Figure 4: Temporal pooling effects. We set $T = 32$ and $k_w = d_w = k_h = d_h = 3$, varying the temporal kernel size and stride.

Table 6: The image results. ⋆ means self-implementation.

| Model | Resolution | MMMU(val) | MathVista | MMB-ENG | MMB-CN | MM-Vet | SEED-IMG | MME | POPE |
|---|---|---|---|---|---|---|---|---|---|
| LLaVA-1.5-13B | 336*336 | 36.4 | 27.6 | 67.8 | **63.3** | 36.3 | 68.2 | 1531/295 | 85.93 |
| LLaVA-Next-7B | 672*672 | 35.8 | 34.6 | 67.4 | 60.6 | 43.9 | 70.2 | 1519/332 | 86.53 |
| VideoLLaVA | 336*336 | - | - | 60.9 | - | 32.0 | - | - | 84.40 |
| Chat-Univ-1.5 | 336*336 | - | - | 62.7 | - | 28.3 | - | - | 85.40 |
| LLaVA-Next-Video ⋆ | 336*336 | 34.2 | 28.9 | 64.7 | 56.7 | 44.0 | 64.6 | 1501/**351** | 83.10 |
| PPLLaVA | 336*336 | **37.9** | **34.6** | **68.9** | 62.0 | **44.7** | **70.7** | **1539**/277 | **88.46** |

and thanks to the prompt-guided approach, the performance remains almost unaffected. In contrast, as shown in Fig. 4, pooling in the temporal dimension yields smaller efficiency gains compared to spatial scaling, with more noticeable performance degradation as the kernel and stride sizes increase. When the pooling kernel and stride are large, the efficiency gains tend to plateau, but the decline in effectiveness becomes significantly pronounced, particularly in spatial pooling, where the performance drop is more severe. Considering all factors, for video input, we ultimately selected a pooling kernel and stride of (2, 3, 3) to ensure a substantial improvement in efficiency while maintaining stable model performance.

**Image Performance.** Theoretically, further video tuning on top of an image-domain LLM could lead to catastrophic forgetting of pre-trained knowledge and image understanding. The PPLLaVA method can also be seamlessly applied to images. Although images do not have the same need for token compression as videos, and compression may lead to performance loss, the guidance from user prompts can still similarly enhance performance. In Table 6, we present PPLLaVA's results on various popular image LLM benchmarks. Since PPLLaVA was trained on LLaVA-1.5 image data based on LLaVA-Next, we compared the results of these two models. We also compare the image performance with LLaVA-Next-Video and other image-video unified models. As shown, PPLLaVA shows a significant advantage in image performance compared to video models, indicating that PPLLaVA has effectively retained pre-trained knowledge. Compared to image models, despite using a smaller LLM or lower image resolution, PPLLaVA, as a video model, still achieved better results on most benchmarks. Notably, our pooling method reduced the visual tokens to one-ninth of the original count at the same resolution. This demonstrates that PPLLaVA can achieve both performance and efficiency improvements even on image-based tasks, highlighting its potential for lightweight multimodal LLM.

## 5 CONCLUSION

In this paper, we propose Prompt-guided Pooling LLaVA (PPLLaVA), a novel pooling method that achieves token compression and prompt-aware feature extraction simultaneously. We first observed that current video LLMs struggle to balance performance on both long and short videos. Further analysis revealed that redundant tokens in videos negatively impact video understanding performance. To address this, our model incorporates three key modules: Fine-grained Vision-Prompt Alignment, Prompt-Guided Convolution-Style Pooling, and CLIP Context Extension. These modules significantly reduce the visual context while effectively extracting essential visual features. Extensive experiments have demonstrated the effectiveness of PPLLaVA on both images and videos, as it achieves the best results across benchmarks of various tasks and video lengths, ensuring excellent efficiency, with particularly outstanding performance on long videos.

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

Table 7: The ablation study on the Pooling Approach. We report the overall performance of VideoMME (w/ subs).

| Pooling Method | kernel1 | kernel2 | tokens | Overall |
|---|---|---|---|---|
| weighted average | (2,3,3) | - | 1024 | **53.6** |
| separate S-T | - | - | 608 | 44.1 |
| max pooling | (2,3,3) | - | 1024 | 52.0 |
| multiple | (1,6,6) | (8,2,2) | 1088 | 52.8 |
| multiple | (4,3,3) | (2,4,4) | 1088 | 53.2 |

Table 8: The study on whether PPLLaVA helps long video understanding. We report the Long performance of VideoMME (w/ subs).

| train | | test | | tokens | Long |
|---|---|---|---|---|---|
| frames | kernel | frames | kernel | | |
| 32 | (2,3,3) | 32 | (2,3,3) | 1024 | 45.7 |
| 32 | (2,3,3) | 64 | (4,3,3) | 1024 | **47.4** |
| 16 | (1,3,3) | 16 | (1,3,3) | 1024 | 43.5 |
| 8 | (1,1,1) | 8 | (1,1,1) | 4608 | 41.2 |

Table 9: The study on multimodal data with interleave training and DPO training.

| Model | Video | Image | Multi-Image | Interleave | DPO | VcgBench | MvBench | VideoMME |
|---|---|---|---|---|---|---|---|---|
| LLaVA-Next-Video | ✓ | ✓ | | ✓ | | 3.26 | - | - |
| | ✓ | ✓ | | ✓ | ✓ | 3.66 | - | - |
| PPLLaVA | ✓ | | | | | 3.20 | 55.0 | 48.9 |
| | ✓ | ✓ | | | | 3.09 | 49.8 | 44.1 |
| | ✓ | ✓ | | ✓ | | 3.32 | 57.1 | 50.0 |
| | ✓ | ✓ | ✓ | ✓ | | 3.21 | 59.2 | 53.6 |
| | ✓ | ✓ | | ✓ | ✓ | 3.73 | 55.8 | 49.3 |

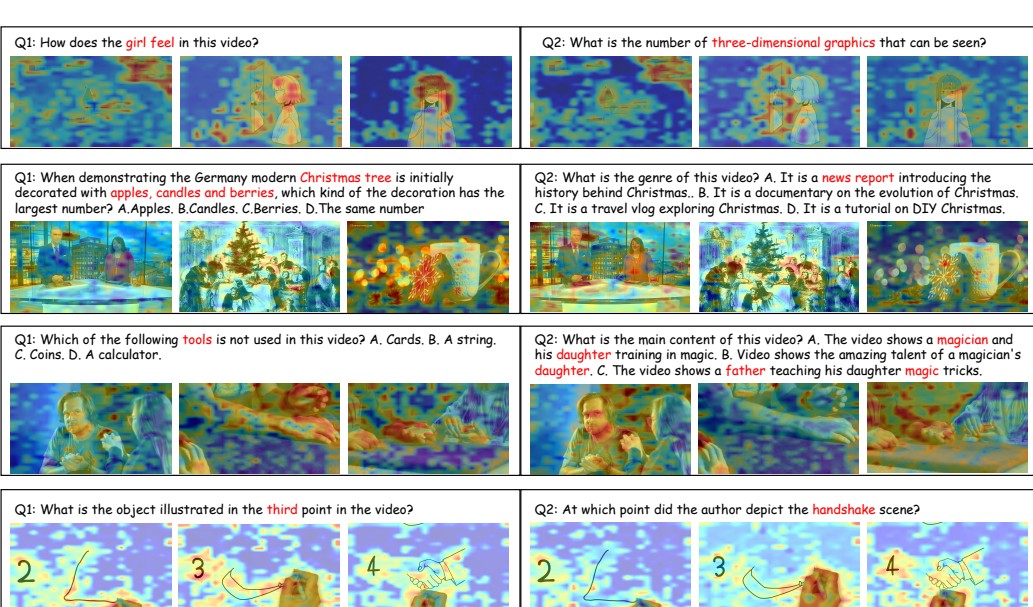

Figure 5: The visualization of the attention weights used to guide video pooling.

## A    MORE ANALYSIS

**Prompt-Guided Pooling Approach.** Beyond the weighted average pooling detailed in the main text, we experimented with several alternative pooling methods guided by the prompt. First, we applied separate spatiotemporal pooling, conducting pooling operations independently on the temporal and spatial dimensions before concatenation. We also explored combinations of different pooling sizes to assess their impact. Lastly, we implemented max pooling using weights derived from the prompt as guidance. As shown in Table 7, spatiotemporal separate pooling demonstrates the worst performance, underscoring the importance of maintaining the 3-dimensional spatiotemporal structure during pooling. Max pooling, though slightly better, still falls short, suggesting that a few prominent features are insufficient to represent the entirety of the video. The combination of various pooling kernels performs similarly to direct weighted averaging when the context length is comparable. Consequently, we opted for weighted averaging, as it provides optimal results while maintaining a simpler structure.

**Is PPLLaVA Really Helpful for Long Video?** Token compression is a key feature of PPLLaVA, primarily aimed at enhancing the understanding of long videos. To validate PPLLaVA's effectiveness

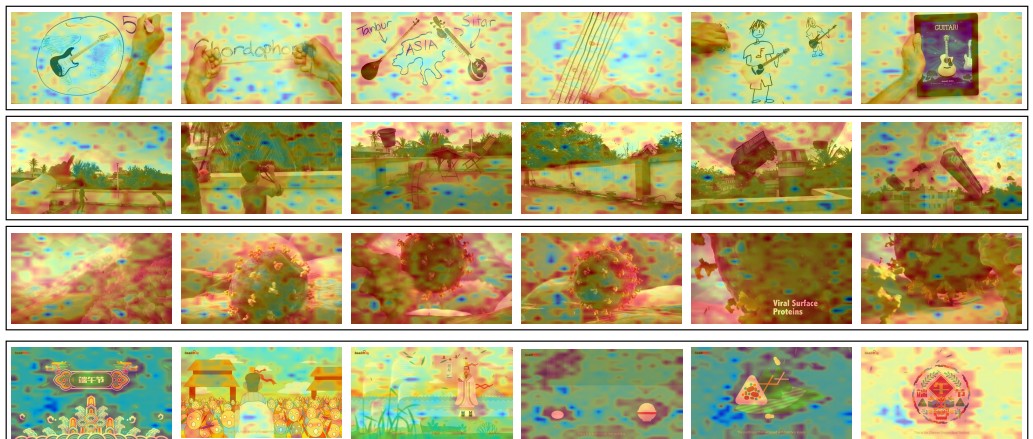

Figure 6: The visualization of the attention weights used to guide video pooling. The prompts for all videos are "Describe this video.".

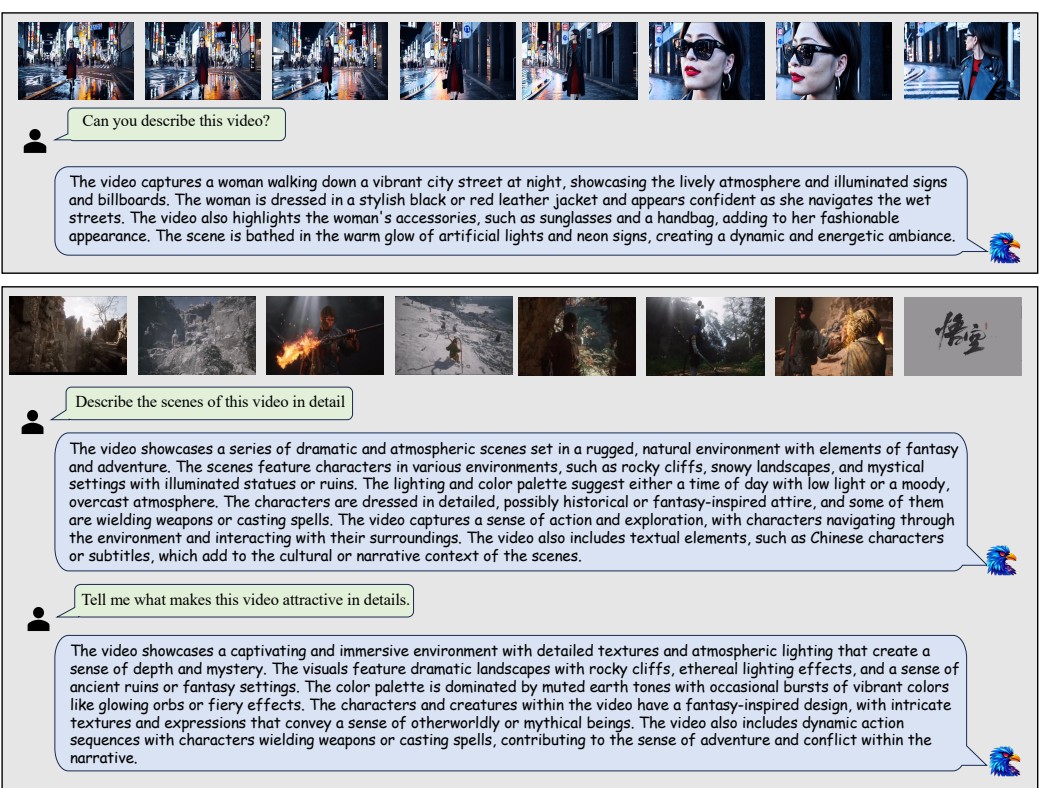

Figure 7: Qualitative result of video summary and detailed video description.

in this regard, we conducted an additional ablation experiment. As shown in Table 8, we first experimented with larger pooling kernels to accommodate more frames. The results indicate that even with some discrepancies between training and testing, using a greater number of frames still improves long video comprehension. When fewer frames are used during training, the disadvantage in long video understanding becomes even more pronounced. In the most extreme case, when no pooling is applied, even with significantly longer context lengths, the understanding of long videos is the weakest due to the limited number of frames that can be accommodated.

**Interleave training and DPO.** Multimodal data interleaved training and DPO are two key techniques utilized in this work. We conducted an analysis of their effects and compatibility with

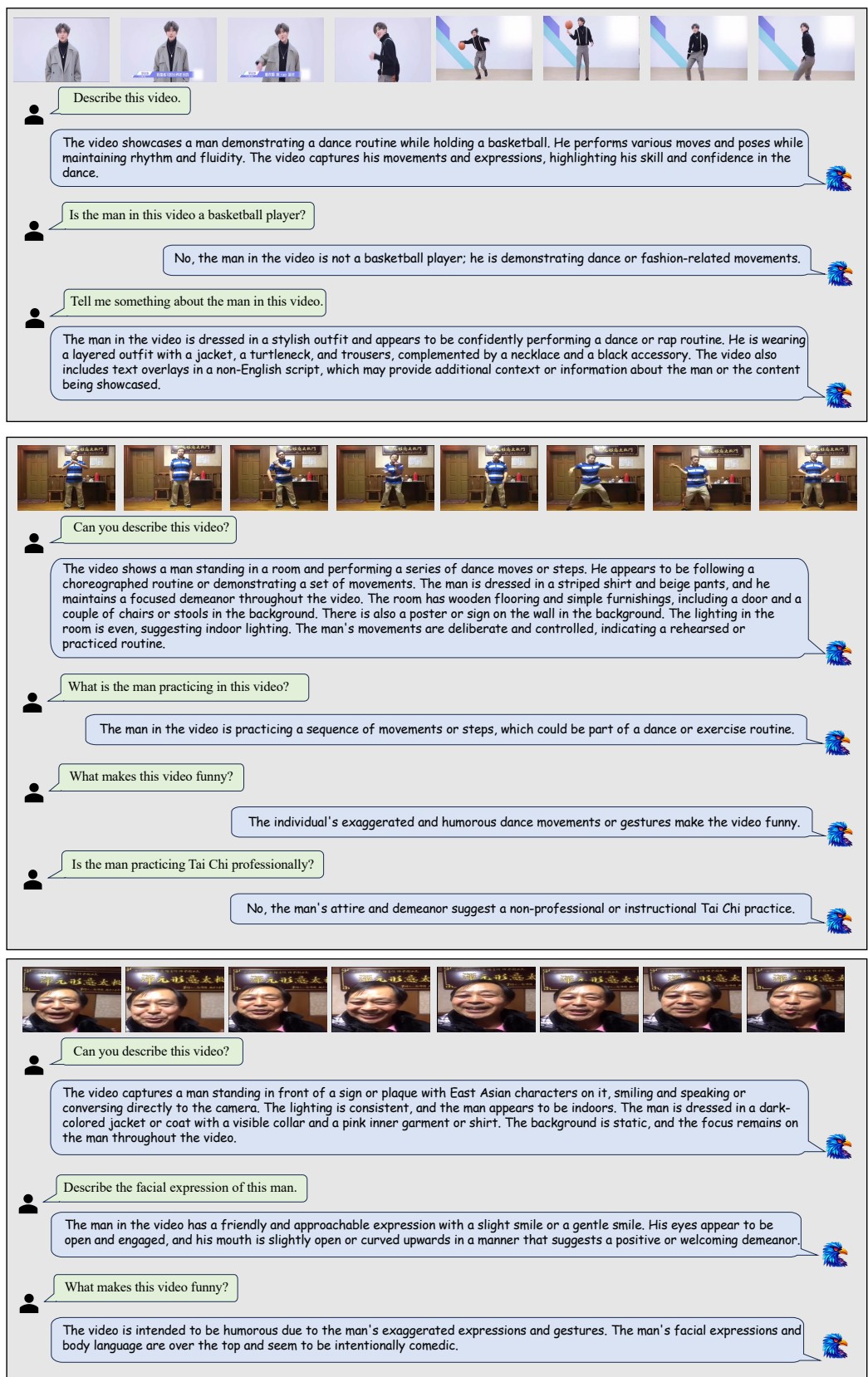

Figure 8: Qualitative result of multi-turn video conversation and reasoning.

PPLLaVA. As shown in Table 9, when different data modalities are not mixed within a batch, adding image or multi-image modalities leads to performance degradation compared to pure video training. This aligns with the conclusions of LLaVA-Next-Video. When data modalities are mixed within the same batch, the additional images enhance performance. However, we found that when further adding multi-image data, the performance on the multiple-choice benchmark improved, but the performance on the caption generation benchmark declined. This indicates that multi-image data can enhance the model's visual knowledge but may reduce its capability in video-based dialogue. In contrast, DPO training has a minimal side effect on multiple-choice benchmarks but significantly improves results in GPT-based evaluation. This highlights DPO's ability to effectively reduce hallucinations in LLM outputs, leading to higher-quality dialogues. Moreover, when compared to the baseline and LLaVA-Next-Video, the combination of DPO and PPLLaVA yields similar improvements. This emphasizes the strong compatibility between PPLLaVA and DPO.

## B  QUALITATIVE RESULTS

In Fig. 5, we visualize the attention weights used to guide video pooling based on the user prompts. For the same video, we tried different questions. It can be clearly observed that the model's attention shifts noticeably depending on the question. For example, when the user asks about the girl's feelings, the attention is significantly focused on her face. Conversely, when asked about the number of 3D objects in the video, the attention shifts more toward the 3D objects. These visualizations demonstrate that while reducing the visual context, PPLLaVA effectively captures the key information in the video. In Fig. 6, we additionally illustrate the attention weights for captioning-related questions, as these questions theoretically provide less informational content. As shown in the figure, prompts like "Describe this video," which lack specific references, result in attention weights being evenly distributed across the foreground. This indicates that our model still plays a significant role in handling captioning-related questions. In Fig. 7 and 8, we further present some examples of video dialogue. As shown in Fig. 7, for the famous Sora video, PPLLaVA can accurately and intricately describe details about the protagonist and the environment. For the more complex scene changes in the trailer for Black Myth Wu Kong, PPLLaVA remarkably captures the details of each scene and character. In Fig. 8, PPLLaVA maintains accuracy and consistency across multiple rounds of dialogue and is capable of making reasonable inferences on open-ended questions.

## C  LIMITATION

Although the 7B PPLLaVA has demonstrated impressive performance, even rivaling that of 34B video LLMs, our biggest regret is that, due to a lack of computational resources, we were unable to train larger-scale LLMs to uncover the limits of this architecture. Additionally, the conflict between the enhanced understanding capabilities brought by multi-image data and the decline in dialogue abilities remains unsolved in this work; a reasonable data allocation ratio might address this issue. We leave these problems for future work.

