# OpenReview forum: "PPLLaVA: Varied Video Sequence Understanding With Prompt Guidance"
_ICLR.cc/2025/Conference — Submitted to ICLR 2025_

### Official Review · Reviewer_Qkbp · 2024-11-03

**Soundness:** 3
**Presentation:** 3
**Contribution:** 2
**Rating:** 3
**Confidence:** 4

**Summary:**

The paper introduces a novel approach to video understanding by leveraging prompt guidance to enhance the performance of video language models (VLMs). The methodology focuses on reducing video redundancy and extracting key content to enhance the performance of VLMs. It uses CLIP-based visual-prompt alignment to extract relevant visual information and compresses visual sequences using convolution-style pooling. Direct Preference Optimization (DPO) was also used to improve its performance. In experiments, PPLLaVA demonstrated good results on both long and short video benchmarks, and achieves over an 80% compression rate.

**Strengths:**

1. The paper introduces methods for video understanding by leveraging prompt guidance through the use of CLIP-based visual-prompt alignment and convolution-style pooling.
2. PPLLaVA shows versatility by performing well on both long and short video benchmarks, demonstrating robust performance for varied video sequence understanding. This adaptability is crucial for handling diverse video lengths and complexities, making it a robust solution for varied video sequence understanding5.
3. PPLLaVA uses Direct Preference Optimization (DPO) to reduce hallucinations in video-based dialogue and also applies CLIP Context Extension to expand text encoding capacity.

**Weaknesses:**

1. PPLLaVA does not show leading results compared to some training-free compression methods like SLOWFAST-LLAVA[1], on benchmarks such as MSVD and MSRVTT.
2. The use of Direct Preference Optimization (DPO) and Proximal Policy Optimization (PPO) lacks innovation. Also, LLaVA-Next-Video also achieve great accuracy improvements using above methods, but this paper does not highlight any unique advantages of these methods within PPLLaVA.

[1] SlowFast-LLaVA: A Strong Training-Free Baseline for Video Large Language Models

**Questions:**

1. The method seems similar to the involution kernel[1]. What are the  differences between the two?
2. Have any ablation studies been conducted using regular 3D convolution for pooling instead of prompt-guided methods?
3. In Table 5, why is the context length for average pooling set to 576 instead of 1024?
4. Can PPLLaVA  extend to multimodal **generative** models?

[1] Involution: Inverting the Inherence of Convolution for Visual Recognition

---

> ### Author Response · Authors · 2024-11-15
> **Response to Reviewer Qkbp**
>
> Thank you for the comments. We are shocked and disheartened by the reviewer's reasons for rejecting our paper. We believe there exist some misunderstandings. We have addressed all of your concerns point by point below. It is appreciated if you have any further feedback on our response.
> ### Q1:  PPLLaVA does not show leading results compared to some training-free compression methods like SLOWFAST-LLAVA, on benchmarks such as MSVD and MSRVTT.
> （1）We use the GPT-3.5-turbo version 0613, which is consistent with previous works like LLaVA-Next-Video, ST-LLM, and Chat-UniVi, while SlowFast-LLaVA uses version 0125. The version of GPT has a significant impact on test results, and version 0125 is more friendly to open-ended VideoQA. To demonstrate this, we re-tested our MSVD and MSRVTT answers using version 0125, and the results are as follows:
> | Method   | context|MSVD-QA|MSRVTT-QA|Anet-QA|
> |:--------|:------:|:------:|:------:|:------:|
> | SlowFast-LLaVA-7B   |3680| 79.1/4.1 | 65.8/3.6 | 55.5/3.4|
> | PPLLaVA-7B   |1024| 79.5/4.2 | 66.9/3.8|56.7/3.6 |
>
> As seen, with the same GPT version, we achieved better results than SlowFast-LLaVA with higher efficiency. Additionally, multiple-choice question answering can mitigate fairness issues related to GPT evaluation. Unfortunately, SlowFast-LLaVA lacks results from more specialized Video LLM benchmarks like MvBench and VideoMME.
>
> （2）MLLMs have numerous benchmarks, and it is unfair to dismiss an entire model simply because it does not achieve leading results in one or two benchmarks. For example, LLaVA-Next-Video performs poorly on MvBench, but we would not dismiss LLaVA-Next-Video based on that alone. It's essential to assess models comprehensively across multiple benchmarks of different kinds.
>
> ### Q2:  The use of Direct Preference Optimization (DPO) and Proximal Policy Optimization (PPO) lacks innovation. Also, LLaVA-Next-Video also achieve great accuracy improvements using above methods, but this paper does not highlight any unique advantages of these methods within PPLLaVA.
> We have never claimed that the innovative use of DPO is our main contribution in the paper. Rather, what we want to emphasize is the compatibility of PPLLaVA with DPO (Line 701), and the richer choices provided by our codebase (Line 119). As can be seen:
> | Method   | VCGBench-avg (w/o DPO)|VCGBench-avg (w/ DPO)|Improvement|
> |:--------|:------:|:------:|:------:|
> | LongVA  |3.26| 3.58 | +0.32|
> | LLaVA-Next-Video  |3.26| 3.66 | +0.40|
> | PPLLaVA   |3.32| 3.73 | +0.41|
>
> PPLLaVA + DPO achieved the largest growth. This demonstrates the good  adaptability between PPLLaVA and DPO. But in any case, this is just a secondary aspect of the paper. We argue that rejecting our paper simply because the subordinate content "lacks innovation" is unfair.
>
> ### Q3: Difference with Involution
> Traditional convolutions share kernels in the spatial dimensions, while Involution shares kernels across channels. In PPLLaVA, the kernel used during pooling is not shared spatially, which indeed makes it similar to Involution. However, PPLLaVA does not have spatial-specific features. Therefore, PPLLaVA employs an operation similar to convolution, but it is neither a true convolution nor Involution.
>
> ### Q4: Have any ablation studies been conducted using regular 3D convolution for pooling instead of prompt-guided methods?
> Using regular 3D convolution without prompt guiding is equivalent to `nn.AdaptiveAvgPool3d`, which is essentially the content of PLLaVA. This comparison with PLLaVA is already provided in the original paper.
>
> ### Q5: In Table 5, why is the context length for average pooling set to 576 instead of 1024?
> The meaning of average pooling is to directly pool the temporal dimension (T) into 1, so the context length is equal to the image context length. In LLaVA-Next, a single frame image is encoded into 24x24, which results in 576 tokens.
>
> ### Q6: Can PPLLaVA extend to multimodal generative  models?
> Sorry, we are not experts in the generation field. We speculate that PPLLaVA could be applied to video editing. By inputting an editing prompt and the corresponding video, PPLLaVA could provide more accurate and generalized editing text for diffusion model. Due to PPLLaVA’s efficiency, it would not significantly increase the editing time.

---

> ### Author Response · Authors · 2024-11-23
> **Looking forward to a reponse before the deadline**
>
> Dear Reviewer Qkbp,
>
> We want to send you a friendly reminder that the stage of discussion will be completed soon.
>
> It is greatly appreciated if you are willing to reconsider your score based on our responses, and we really want to know whether our responses address your concerns. If there is any other concern that we could not address in the response, please feel free to let us know and we would be happy to provide further explanation.
>
> With sincere regards,
>
> Authors of Paper 1865

---

> > ### Comment · Reviewer_Qkbp · 2024-11-27
> >
> > Thanks for your reply, I'll maintain my score.

---

> > > ### Author Response · Authors · 2024-11-27
> > > **Response to The New Comment**
> > >
> > > Thank you for your response to our rebuttal. However, we must express our concern that your feedback, specifically your decision to maintain your score with a brief "Thanks for your reply, I'll maintain my score," does not provide sufficient insight into the specific issues you may still have with our work.
> > >
> > > As a responsible reviewer, we believe it is crucial to offer more detailed feedback, especially after we have put significant effort into addressing your concerns. If you feel that our revisions have adequately addressed your points, we would appreciate a more thorough explanation of why the score remains unchanged. Alternatively, if the issues have not been fully resolved, we kindly request a clear and specific outline of the remaining problems so that we can continue to improve the paper.
> > >
> > > A brief, unelaborated reply is not only unhelpful but also unconstructive to the process of academic review. We sincerely hope you can provide more substantial feedback to help guide us in finalizing our work.
> > >
> > > We look forward to your more detailed input.
> > >
> > > Best regards,

---

### Official Review · Reviewer_vFqm · 2024-11-04

**Soundness:** 3
**Presentation:** 2
**Contribution:** 2
**Rating:** 6
**Confidence:** 4

**Summary:**

This paper proposed PPLLAVA, a video-LLM based on a prompt-guide pooling strategy. The core idea is to conduct text query/prompt-dependent pooling for video features before putting them into LLM. The authors conduct extensive experiments to show the effectiveness of the proposed method. The proposed PPLLAVA outperforms other training-free/training-based video-LLM at a similar model scale.

**Strengths:**

(1) this paper conducts extensive experiments to show the effectiveness of the proposed methods.

(2) pooling redundant tokens based on visual-text similarity is an efficient solution to capture query-related information in a video with large redundancy.

(3) The design of extending the CLIP content window is smart enough to adapt CLIP to a longer context with minimal modification.

**Weaknesses:**

(1) one thing I'm a bit confused about is the video captioning setting when adapting this prompt-guided pooling strategy. As the caption prompts are not as diverse as that question answering, holistic captioning should be affected unless the user is querying a specific object-related caption. I would suggest some experiments on video caption benchmarks, e.g. activitynet caption or newer DREAM-1K, to further support the strong video understanding ability claim.

(2) it seems the paper missed some related baseline methods, see LongVA [1], and Kangaroo [2]. It should be helpful to include a more comprehensive analysis and comparison with those methods, even if some of them achieve better performance on some datasets.

(3) My other big concern is the novelty side. While I understand this paper did a good job of validating their idea/design with extensive experiments and ablations, comparing some zero-shot baseline like PLLAVA/LLaVA-NeXT-Video, the big boost of the performance might come from fine-tuning on 1.3M multimodal data and the prompt-guided pooling seems a bit incremental from my personal perspective. So I am a bit worried that if we conduct the same level of training with a stronger LLM backbone that has a long content window, the model can learn this query-related attention itself, so the proposed core idea, prompt-guided pooling, seems sub-optimal.

(4) A potential solution might be to fine-tune zero methods that conduct their token merging idea with the same data and show the performance difference.

[1] Long Context Transfer from Language to Vision
[2] Kangaroo: A powerful video-language model supporting long-context video input

**Questions:**

Please see the weaknesses.

---

> ### Author Response · Authors · 2024-11-15
> **Response to Reviewer vFqm (The First Part)**
>
> Thank you for the valuable comments. We are sorry for some misunderstandings on our paper. We believe that our response can help clarify and address some potential misunderstandings. We have addressed all of your concerns point by point in two pages. It is appreciated if you have any further feedback on our response.
>  ### Q1: Confusion about Video Caption.
> We agree with the reviewer’s point that the caption prompts provide limited information, which may affect the method's performance. In fact, both our training and test sets contain a large number of caption samples. For example, in the VideoChatGPT-Bench, over 50% of the questions are caption-style prompts such as "What is happening in the video?" and "What is the sequence of events in the video?". Despite this, our method still performs well on VideoChatGPT. We speculate that when encountering holistic captioning prompts, our method might sample the entire video more uniformly and even adaptively extract video features that are helpful for summarizing the video (due to the large number of caption training samples).
>
> We understand that this may not fully address the reviewer's concern. Therefore, following the reviewer's suggestion, we tested our model on the ActivityNet caption. Due to the lack of comparable methods, we only compared our model with our Average Pooling baseline and the AdaptivePooling3D without CLIP guidance. The testing process followed the same procedure as ActivityNet-QA using GPT evaluation (we adopted the score because the accuracy of captioning is difficult to measure). The results are as follows:
> | Method   |Context| ActivityNet-Caption-score|
> |:--------|:------:|:------:|
> | Average Pooling   | 576 |  2.87 |
> | AdaptivePooling3D   |  1024 |  3.12 |
> | Prompt-Guided Pooling   |  1024 | 3.22|
>
> The results show that our Prompt-Guided Pooling method still outperforms AdaptivePooling3D, indicating that our approach remains effective for captioning tasks. However, we must acknowledge that the improvement is not as large as observed in other datasets, suggesting that the limited information provided by caption prompts does have some impact on PPLLaVA's performance.
>
>  ### Q2: Comparison with LongVA and Kangaroo.
> Thanks for the reminder. The reason we didn't make the comparison earlier is that our model was based on Vicuna-7B, while LongVA and Kangaroo used more powerful LLMs. To ensure a fair comparison, we retrained our model based on LLaVA-Qwen2. The same as the experiment in Reviewer 1Y1n’s Q1.2, we trained on our 600k corpus (300k LLaVA-1.5 image data + LLaVA-Hound-300k video data, totaling 600k) with 64 frames per video. The results are as follows:
>
>  | Method   | LLM | SFT Data | Context | VCGBench-avg  |VideoMME-overall|VideoMME-Long|
> |:--------|:------|:------:|:------:|:------:|:------:|:------:|
> | LongVA  | Qwen2-7B   | 760k    | 224k | 3.58 | 52.6 |46.2|
> | Kangaroo   |  Llama-3-8B-Instruct  | 2.24M    | 22k | - | 56.0 |46.7| 53.5|
> | LLaVA-OneVision   |  Qwen2-7B  | 4.8M    | 8k | 3.51 | **58.2** |46.7|
> | PPLLaVA   | Qwen2-7B   | 600k    | 2k | **3.63** | 55.2 | **47.1**|
>
> As can be seen, under the new baseline, PPLLaVA still achieves very competitive results. Despite the disadvantages in context length and training data, PPLLaVA achieves the optimal result on VCGBench, while also demonstrating comparable performance on Video-MME. Notably, our method still achieves the highest result on Video-MME-Long, indicating that PPLLaVA not only maintains the highest efficiency and fastest throughput, but also excels in long video understanding performance.

---

> > ### Comment · Reviewer_vFqm · 2024-11-25
> > **Thanks for your rebuttal**
> >
> > Q1: Thanks authors for providing more caption data details to address my confusion. I appreciate the extra experiments on different pooling operation, it shows the effectiveness of your method. However, the claim
> > ```
> > We speculate that when encountering holistic captioning prompts, our method might sample the entire video more uniformly and even adaptively extract video features that are helpful for summarizing the video.
> > ```
> > is still not grounded and lacks support. I would appreciate it if the authors could provide more thoughts on this.
> >
> > Q2: thanks for providing those new comparisons with Kangaroo and LongVA. Additional questions: Why does context length vary across the same LLM backbone in this table?

---

> > > ### Author Response · Authors · 2024-11-25
> > > **Thanks for the first response！**
> > >
> > > We are very grateful to the reviewer for being the first to respond to us. In the following replies, we address each of your new concerns point by point. If your concerns have been addressed, we respectfully and politely request for a reconsideration of the score. Thank you once again for your time and consideration. We look forward to hearing from you.
> > >
> > >  ### Further response for Q1
> > > Honestly, the aforementioned hypothesis is difficult to quantitatively verify beyond the good performance on caption datasets. Therefore, to validate our idea, we provided some qualitative evidence. Since it is not possible to include figures here, we have updated our manuscript. In the new manuscript, **Figure 6** visualizes the attention weights used to guide video pooling for captioning-related questions. As shown in the figure, despite the lack of sufficient information in prompts like "Describe this video," the attention weights still manage to distribute relatively evenly across the foreground, filtering out some meaningless background. This indicates that our model still plays a role in handling captioning-related questions.
> > >
> > >  ### Further response for Q2
> > > Here, the context length refers to the visual sequence used in the context. LLaVA-OneVision has essentially reached the original limit of the LLM, while LongVA employs its own unique method for extending the context. We used a shorter context length for two reasons: first, fewer contexts mean higher efficiency, which is one of PPLLaVA's advantages. Second, our computational resources are indeed limited. LLaVA-OneVision utilized over 64 A100-80G GPUs for training, whereas we only have 8. If we were to use an 8K context length, it would require an extremely high accumulation. Being temporarily unable to explore the performance ceiling of PPLLaVA is both a limitation and a regret on our part.

---

> > > > ### Comment · Reviewer_vFqm · 2024-11-25
> > > > **Thanks for your response**
> > > >
> > > > Thanks authors for providing extra answers to my question.
> > > >
> > > > Q1: I agree quantitative evaluation/metrics would be hard. The explanation of the intermedium attention map partially makes sense to me, but it still lacks clear motivation.
> > > >
> > > > Q2: Thanks for your extra clarification. It is clear to me now.
> > > >
> > > >
> > > > Overall, I believe this paper's experiments for effectiveness are enough. I believe there is no reason for me personally to reject this paper, and **I will also raise my score to 6**.
> > > >
> > > > However, I also need to state that even though I acknowledge the effectiveness and design of this paper, I still hold the personal view that the novelty of the paper is limited, and the reason why ppllava works is still unclear to me.  I acknowledge this implementation, but it doesn't seem necessary for advanced VLM, and it is even a little complex for me.

---

> > > > > ### Author Response · Authors · 2024-11-27
> > > > > **Response to Reviewer vFqm**
> > > > >
> > > > > Thank you for raising your score! Even though it is nearly impossible to improve the score at this point, we still want to provide further clarification for your concerns. We once again thank the reviewer for their kind response.
> > > > >  ### Why ppllava works on captioning.
> > > > >  We answer this question from a different perspective. While the context of LLMs is long, some works [1] have shown that the "effective context" of LLMs is actually quite limited, with the earlier parts being more effective than the later parts. In our model, even if the prompt is neutral, it can at least filter out black screens and solid-color backgrounds, which are common redundant content in videos. Therefore, the performance improvement in PPLLaVA is not only due to prompt-guidance, but also to compression—by compressing more valuable content into the more precious front-end context.
> > > > >   ###  It doesn't seem necessary for advanced VLM, and it is even a little complex for me.
> > > > > In fact, the implementation of PPLLaVA is not complicated. For an MLLM, we only need to modify its **multi_modal_projector** part. Therefore, we can quickly implement it on new baselines like LLaVA-Qwen.
> > > > >
> > > > > On the other hand, we acknowledge that our approach may not be the most elegant for training foundational models like LLaVA (i.e., complex) . However, if the goal is to achieve a lightweight, high-throughput MLLM, our model is still a strong choice. We can improve performance while achieving an 8x throughput increase.
> > > > >
> > > > > [1] RULER: What's the Real Context Size of Your Long-Context Language Models?

---

> ### Author Response · Authors · 2024-11-15
> **Response to Reviewer vFqm (The Second Part)**
>
> ### Q3.1: Novelty Concerns: the big boost of the performance might come from fine-tuning on 1.3M multimodal data and the prompt-guided pooling seems a bit incremental from my personal perspective.
> Aside from the method itself, there are indeed many significant factors that affect performance, such as data and LLM. As shown in Q2, changing the baseline LLM significantly improved performance. However, this does not affect our evaluation of the method's inherent effectiveness.
>  To demonstrate this, we select a few methods from the original manuscript for fair comparison below. They use the same Image LLM, with identical LLM sizes, similar amounts of instruction tuning data, and none of them use multi-image data:
>  | Method   | Pretrain | SFT Data | Context | VCGBench-avg  |VideoMME-overall|MVBench-avg|ActivityNet-QA|
> |:--------|:------:|:------:|:------:|:------:|:------:|:------:|:------:|
> | PLLaVA   | LLaVA-Next-Vicuna7B   | 783k    | 1k | 2.99 | - |46.6| **56.3**|
> | LLaVA-Next-Video   | LLaVA-Next-Vicuna7B   | 860k    | 8k | 3.26 | 45.0 |33.7| 53.5|
> | PPLLaVA   | LLaVA-Next-Vicuna7B   | 760k    | 1k | **3.32** | **50.0** |**57.1**| 56.1|
>
>  As can be observed, without multiimage data, PPLLaVA still demonstrates strong performance across various test sets under fair comparison, with significant advantages on most benchmarks. These advantages are sufficient to demonstrate that our improvements stem from the new pooling technique. Factors beyond the method itself, such as more data and stronger LLMs, will affect all methods, so they do not impact our evaluation of the method's inherent effectiveness.
>
>  ### Q3.2: Novelty Concerns: So I am a bit worried that if we conduct the same level of training with a stronger LLM backbone that has a long content window, the model can learn this query-related attention itself, so the proposed core idea, prompt-guided pooling, seems sub-optimal.
> A stronger LLM backbone and sufficiently long context can indeed solve many problems. For example, the strategies used in LLaVA-Next-Video and LLaVA-OneVision is to directly fill the long LLM context with video tokens, letting the LLM itself interpret the user query and the video. However, this does not conflict with our approach:
>
>  (1) One major advantage of our method is that it can improve performance while enhancing efficiency. Filling the context with video tokens can indeed guarantee performance, but long contexts lead to extremely long throughput times, which is not practical for real-world applications. PPLLaVA ensures good performance while providing fast response times.
>
> (2) No matter how large the LLM’s context is, it can only accommodate a limited number of video frames. Our method, however, allows more video frames to be input into the same context, thus improving long video understanding performance.
>
> (3) Following the reviewer’s suggestion, in Q2, we fine-tuned our method on a stronger LLM. As seen, our method still achieves good results with the more powerful LLM, demonstrating the versatility of our approach and idea.

---

> > ### Comment · Reviewer_vFqm · 2024-11-25
> > **Thanks for your rebuttal**
> >
> > Q3.1: Thanks for providing extra clarifications. I'm almost stratified with your answer to my question on training data though it is not a totally apple-to-apple comparison.
> >
> > Q3.2: Thanks for your extra discussion on the method's effectiveness. I'm almost stratified with your answer.

---

> ### Author Response · Authors · 2024-11-23
> **Looking forward to a reponse before the deadline**
>
> Dear Reviewer vFqm,
>
> We want to send you a friendly reminder that the stage of discussion will be completed soon.
>
> It is greatly appreciated if you are willing to reconsider your score based on our responses, and we really want to know whether our responses address your concerns. If there is any other concern that we could not address in the response, please feel free to let us know and we would be happy to provide further explanation.
>
> With sincere regards,
>
> Authors of Paper 1865

---

### Official Review · Reviewer_hDF1 · 2024-11-04

**Soundness:** 3
**Presentation:** 3
**Contribution:** 3
**Rating:** 3
**Confidence:** 5

**Summary:**

The paper proposes a novel method named PPLLAVA, which is based on Video LLM. my concerns are as below.
1. The authors claim that the method is for long video, however, I cannot see the comparison of performances when applying the method to long/short videos. Also, how to define "long video"? 1 hour long?
2. I think LLAVA is for images, instead of video. While if the authors want to apply LLAVA to videos, why not do like this: a. locate the related frames; b. VQA on frames. I cannot see the necessity of the three steps way.
3. The authors seems want to satisfy the user's needs/reply to user's questions. What if the questions are open-ended questions without an answer?
4. why we need "Prompt-Guided Pooling"? I cannot see the necessity of prompt here.

**Strengths:**

The paper is well-written, with clear pipeline, framework, and experimental results.

**Weaknesses:**

As can be found in the summary. I cannot see why the approach should be applied to videos instead of image-level applications.

**Questions:**

Will the method also be applied to image-level applications? is video necessary?

---

> ### Author Response · Authors · 2024-11-15
> **Response to Reviewer hDF1**
>
> To be honest, it was difficult to remain calm after reading the reviewer's comments. So many of the issues have been mentioned multiple times in the paper, yet the reviewer still writes "I cannot see". Regardless, we appreciate the reviewer's efforts. We have addressed all of your concerns point by point below. We sincerely hope that the reviewer will take the review process seriously, so that the efforts of everyone involved are not wasted.
>   ### Q1: I cannot see why the approach should be applied to videos instead of image-level applications. Will the method also be applied to image-level applications? is video necessary?
>  （1）We have emphasized multiple times in the paper! Videos, especially long videos, introduce a large number of tokens, while the LLM's context can only accommodate a limited number of video frames, which necessitates compressing the video tokens. On the other hand, a single image has a limited number of tokens, so models after LLaVA 1.5, by cutting the grid, even increase the number of tokens to achieve better visual understanding. Therefore, PPLLaVA is more needed in videos than in images.
>
>  （2）We do have PPLLaVA's results on images! And we even mentioned this in the abstract!!! The image results can be found in Table 6 of the main text. After conducting video instruction tuning, PPLLaVA did not compromise image performance; on the contrary, it achieved better results with higher efficiency.
>
>  ### Q2: I cannot see the comparison of performances when applying the method to long/short videos. Also, how to define "long video"? 1 hour long?
> (1) The comparison between long and short videos is already provided in Table 3 of the main text!!! In Table 8, we have also conducted an additional ablation study on whether PPLLaVA contributes to long video understanding!
>
> (2) Most of the benchmarks in the Video LLM field consist of short videos ranging from a few seconds to a few minutes. However, in VideoMME, the "long" video duration can reach 30 minutes to 1 hour, which can absolutely be considered long video within the field.
>
>  ### Q3: I think LLAVA is for images, instead of video. While if the authors want to apply LLAVA to videos, why not do like this: a. locate the related frames; b. VQA on frames. I cannot see the necessity of the three steps way.
>  Almost all Video LLM papers are built upon image-domain LLMs like LLaVA. Before forming the idea, we "think" the reviewer is expected to at least read one or two papers in our field.
>
> ### Q4: What if the questions are open-ended questions without an answer?
> In training, questions without answers are not included. During testing and real-world usage, if the questions are open-ended questions without an answer, PPLLaVA would try to provide the most reasonable response depending on the question and video content. What if the questions are open-ended questions without an answer for ChatGPT?
>
> ### Q5: why we need "Prompt-Guided Pooling"? I cannot see the necessity of prompt here.
> We've already addressed this point countless times in the paper!!!!  Prompt=user question=user instruction in our context. Direct pooling often leads to performance degradation, whereas pooling while extracting video content relevant to the user’s question helps avoid this performance loss.

---

> > ### Comment · Reviewer_hDF1 · 2024-12-03
> > **The problems are not solved.**
> >
> > Apologies for the delayed response; the end of the year has been quite busy.
> >
> > I believe the author did not fully comprehend the questions, and their responses did not address my concerns. My brief understanding of the paper is that the author attempted to introduce pooling at the video level and reduce the context token, thereby enabling LLMs to process so-called hour-long videos. However, this task is relatively straightforward and can be accomplished with basic video summarization combined with LLAVA. Additionally, in video summarization, most videos exceed 10 hours in length, making it difficult to technically define "long" without a reference.
> >
> > Extracting content based on user instructions or queries, followed by question answering, constitutes a standard RAG framework and is not novel.
> >
> > Returning to the topic of video, the paper "Intentvizor: Towards generic query guided interactive video summarization" demonstrates summarizing videos according to user needs (queries), which is essentially what the current paper claims. This paper also employs the slow-fast feature, similar to the current paper. Therefore, summarizing videos based on user instructions or queries is not a new concept but rather a traditional approach in video summarization. Consequently, prompt-guided pooling is unnecessary. This is the primary reason for my unchanged score.
> >
> > Regarding Q3, not all video understanding papers are based on LLAVA. If the authors intend to use LLMs as a baseline, GPT-4o would likely yield better performance and serve as a more robust baseline.
> >
> > For Q4, addressing the practical issue of open-ended questions without answers for ChatGPT, the current model evaluates such questions from various perspectives, including groundedness, relatedness, and correctness, and assigns scores. The authors should possess relevant knowledge before applying LLMs and using LLAVA.
> >
> > Additionally, more practical scenarios arise when the query lacks specific instructions.
> >
> > Regarding Q5, as previously mentioned.
> >
> > Based on the above explanation, I will maintain my score.

---

> > > ### Author Response · Authors · 2024-12-03
> > > **The response once again demonstrates that the reviewer has hardly read our paper.**
> > >
> > > We appreciate the reviewer's final response, even though it came during a classic time period when the authors have little chance to discuss.
> > > 1. The reviewer has consistently emphasized video summarization, seemingly that all your knowledge of Video VLM is limited to the video summarization task. However, video summarization on datasets like SumMe and Shot2Story20K is only a small subtask within Video VLM and has almost no relevance to our paper. Moreover, well-known benchmarks such as VideoMME and LongVideoBench provide widely accepted definitions of long videos. We are puzzled as to why the reviewer insists on using experience from this minor task to criticize our work and dismiss other industry-recognized facts.
> > > 2.  Anyone who has read our paper would not think that our method has any relation to "RAG." Ironically, we would actually prefer if there were a connection, because firstly implementing RAG in Video LLM would be a big novelty! :)
> > > 3. The reviewer insists that using LLaVA alone is sufficient for LLM-based video understanding, and suggests that simply using GPT-4 would be better. According to this logic, all the well-known works such as VideoChat, VideoChatGPT, Chat-UniV, VideoLLaVA, LLaMA-ViD, and LLaVA-Video might be deemed unnecessary. Perhaps, within the narrow scope of video summarization, LLaVA truly can solve all the problems!
> > > 4. PPLLaVA and [1] have completely different implementation methods and baselines, sharing only a similar motivation. This 2021 paper still uses GNNs, before GPT-3 even existed. The "traditional approach" is only relevant to your video summarization task and has almost no overlap with our work. Furthermore, our paper has no relation to "slow-fast features," and we are puzzled as to why the reviewer formed this opinion. Perhaps you misunderstood the comments of other reviewers? ：)
> > > 5. It is quite humorous that in our paper, the MSVD, MSRVTT, ActivityNet, and VideoChatGPT benchmarks in Table 2 all involve open-ended questions, which are scored by ChatGPT from multiple dimensions. We misunderstood the reviewer's point because, based on the initial question, "What if the questions are open-ended questions without an answer?", we couldn't figure out the reviewers mean how these quesitons are tested. The reviewer suggested that we should "possess relevant knowledge before applying LLMs and using LLaVA," and we humbly accept this advice. However, we also suggest that the reviewer perhaps take a moment to read the paper before reviewing.
> > >
> > > [1]"Intentvizor: Towards generic query guided interactive video summarization"

---

### Official Review · Reviewer_tF5Y · 2024-11-04

**Soundness:** 3
**Presentation:** 3
**Contribution:** 4
**Rating:** 6
**Confidence:** 5

**Summary:**

This paper proposes a novel pooling method to enhance video large language models (LLMs) by enabling token compression and prompt-aware feature extraction. The approach can be summarized as follows:
- Prompt-Relevant Visual Feature Extraction: The method begins by identifying visual features relevant to the prompt through a fine-grained visual-prompt alignment.
- 3D Token Compression: Leveraging prompt-vision alignment, the authors employ a 3D convolutional kernel to compress tokens to a specified 3D size, adjusting the output stride accordingly.
- Asymmetric Positional Embedding: To further enhance the model's capabilities, the authors introduce asymmetric positional embedding extensions to expand the capacity of text encoding.

The proposed method achieves a significant compression rate and supports both short and long token inputs. In comparison to Q-Former, the authors argue that their approach offers greater flexibility and adaptability.

Extensive experiments across a variety of datasets demonstrate promising performance improvements with this method.

**Strengths:**

- This paper begins with a clear analysis of video LLMs and highlights limitations in existing pooling techniques, effectively motivating the proposed approach.
- The proposed method, which comprises three key components, is technically sound and well-justified, addressing the identified challenges.
- The training strategy is particularly interesting, utilizing detailed video captions as proxies for video content and performing DPO with feedback from the language model serving as a reward signal.
- Experimental results demonstrate substantial performance improvements, supporting the effectiveness of the proposed approach.
- The paper includes detailed analysis, providing insights into the efficiency and performance gains of the model, with evidence suggesting that DPO plays a critical role in achieving the final results.
- The paper is well-written and easy to understand.

**Weaknesses:**

- The diagrams could benefit from more detailed captions to enhance clarity. For example, Figure 2 requires additional context to quickly convey the relationship between [TOK] and the 3D blue rectangle, as this took some time to interpret.
- The fairness of the comparisons is questionable, as many previous works do not utilize DPO in their training. This difference may give the proposed method an advantage, making it challenging to assess its improvements solely based on the new pooling technique.

**Questions:**

- Since the proposed method builds on the image-domain LLaVA, would it be feasible to adapt this approach to existing video LLMs directly? It would be interesting to understand any potential challenges or modifications required for integration with current video-specific models.

---

> ### Author Response · Authors · 2024-11-15
> **Response to Reviewer tF5Y**
>
> Thanks for the valuable comments. We appreciate you giving positive feedback on our paper. All your concerns are addressed point by point. We believe that our response can help clarify and address some potential misunderstandings. It would be appreciated if you have any further feedback on our response.
> ### Q1: Improving the diagrams.
> We appreciate the reviewer for the suggestion. As suggested by the reviewer, in the new manuscript, we have updated Figure 2 by adding more explanations and context to enhance clarity.
> ### Q2: The fairness.
> We understand the reviewer's concern. In the original manuscript, we had marked the results with DPO, while also retaining the results where PPLLaVA did not use DPO on all datasets to ensure a fair comparison. In fact, DPO only significantly improves the model's performance on dialogue tasks. For Video Question-Answering tasks that do not rely on GPT evaluation, DPO does not have much effect.
>
> To further demonstrate the fairness of the comparisons in the paper, we separately compare a few methods below. They use the same Image LLM, with identical LLM sizes, similar amounts of instruction tuning data, and none of them use DPO. The following results are all from the original manuscript:
>
> | Method   | Pretrain | SFT Data | Context | VCGBench-avg  |VideoMME-overall|MVBench-avg|ActivityNet-QA|
> |:--------|:------:|:------:|:------:|:------:|:------:|:------:|:------:|
> | PLLaVA   | LLaVA-Next-Vicuna7B   | 783k    | 1k | 2.99 | - |46.6| **56.3**|
> | LLaVA-Next-Video   | LLaVA-Next-Vicuna7B   | 860k    | 8k | 3.26 | 45.0 |33.7| 53.5|
> | PPLLaVA   | LLaVA-Next-Vicuna7B   | 760k    | 1k | **3.32** | **50.0** |**57.1**| 56.1|
>
> As can be observed, even without DPO, PPLLaVA demonstrates strong performance across various test sets under fair comparison, with significant advantages on most benchmarks. These advantages are sufficient to demonstrate that our improvements stem from the new pooling technique. The use of DPO or additional data (multi-image) is aimed at achieving the best possible results, and we have provided results without these methods or data. Therefore, it does not affect the fairness of the basic comparison.
> ### Q3: Implementation on existing Video LLM.
> That's a very interesting question! Currently, many top-performing MLLMs tend to directly input all video tokens into the LLM, filling up the context, as seen in LLaVA-Next-Video and LLaVA-One-Vision. As a result, PPLLaVA can also be seamlessly implemented on these methods, reducing visual context to increase throughput or support more video frames as input. To test the effect of this approach, we conducted a tiny experiment with LLaVA-Next-Video by initializing PPLLaVA with LLaVA-Next-Video's weights and performing instruction tuning on VideoChatGPT-100k. The results are as follows:
> | Method   | Context | VCGBench-avg  |VideoMME-overall|MVBench-avg|ActivityNet-QA|
> |:--------|:------:|:------:|:------:|:------:|:------:|
> | LLaVA-Next-Video   | 8k | 3.26 | 45.0 |33.7| 53.5|
> | LLaVA-Next + PPLLaVA   |  1k | 3.32 | 50.0 |57.1| 56.1|
> | LLaVA-Next-Video + PPLLaVA   |  1k | 3.22 | 46.9 |49.1| 52.1|
>
> As can be seen, when implemented on Video LLMs, our method significantly improves efficiency without causing a severe drop in performance. In fact, it leads to notable performance improvements on some benchmarks. However, the results are not as strong as those directly obtained with LLaVA-Next. We speculate that this is because Video LLMs have already been trained to fit video instruction data, and further fine-tuning on video instruction data may lead to overfitting or disrupt the knowledge acquired from the original image-text pretraining.

---

> ### Author Response · Authors · 2024-11-23
> **Looking forward to a reponse before the deadline**
>
> Dear Reviewer tF5Y,
>
> We want to send you a friendly reminder that the stage of discussion will be completed soon.
>
> It is greatly appreciated if you are willing to reconsider your score based on our responses, and we really want to know whether our responses address your concerns. If there is any other concern that we could not address in the response, please feel free to let us know and we would be happy to provide further explanation.
>
> With sincere regards,
>
> Authors of Paper 1865

---

> > ### Comment · Reviewer_tF5Y · 2024-11-26
> > **comments**
> >
> > Thank you to the authors for their detailed responses and the additional experiments, which address my initial concerns. While I am leaning positive for this paper, I encourage the authors to carefully address the remaining concerns raised by fellow reviewers to strengthen the submission.

---

> > > ### Author Response · Authors · 2024-12-02
> > > **Response to Reviewer  tF5Y**
> > >
> > > Thank you for your positive feedback. We greatly appreciate your continued support and the time you’ve dedicated to reviewing our work.

---

### Official Review · Reviewer_1Y1n · 2024-11-04

**Soundness:** 3
**Presentation:** 3
**Contribution:** 3
**Rating:** 6
**Confidence:** 4

**Summary:**

The paper proposes a new model that can handle short and long videos with state-of-the-art performance under comparable model sizes. By using DPO's fine-tuning strategy, the proposed PPLLaVA is able to outperform selected baseline models. This is achieved by incorporating three components: the CLIP-based visual-prompt alignment that extracts visual information relevant to the user’s instructions, the prompt-guided pooling that compresses the visual sequence to arbitrary scales using convolution-style pooling, and the clip context extension designed for lengthy prompt common in visual dialogue.

**Strengths:**

1. The paper's motivation is interesting: applying pooling based on the text-frame similarity is reasonable. This is effective on removing redundant video frames conditioned on the user prompt input.
2. The author also proposes methods to handle long user inputs.
3. The final results are promising.

**Weaknesses:**

1. How the capability of CLIP affects the model performance is not discussed. Since the method relay on CLIP to remove redundant video frames, its accuracy could be the bottleneck of the method. Will a stronger text-image matching encoder further improve the model's performance? I believe adding this type pf discussion to the paper will make it stronger.

**Questions:**

Please refer to the weakness section: Will a stronger text-image matching encoder further improve the model's performance?

---

> ### Author Response · Authors · 2024-11-15
> **Response to Reviewer 1Y1n**
>
> Thanks for the valuable comments. We appreciated you giving positive feedback on our paper. All your concerns are addressed below. It is appreciated if you have any further feedback on our response.
> ### Q1.1: How the capability of CLIP affects the model performance is not discussed.
> We agree with the reviewer's point that the text-image matching performance of CLIP determines the upper limit of our model. In fact, we have attempted to break through this bottleneck in the paper. For example, we found that CLIP's 77-token encoding context struggles to handle long questions or multi-turn conversations. Therefore, we have proposed an additional CLIP context extension method. As shown in Table 5, this modification to CLIP significantly enhances performance.
>
> On the other hand, the choice of text-image matching encoder actually depends on the visual encoder used by the image-domain pretrained MLLM. For example, since LLaVA-Next uses CLIP-L14-336, we must use the corresponding aligned text encoder. Otherwise, if we were to use a new text-image matching encoder, we would have to start from scratch with a multi-stage pretraining, which is highly inefficient. As an alternative option, in Q1.2, we use another image-domain pretrained MLLM, LLaVA-Qwen, which employs a more powerful text-image matching encoder, SigLIP.
>
> ### Q1.2: Will a stronger text-image matching encoder further improve the model's performance?
> To address the reviewer's concern, we have also implemented PPLLaVA on LLaVA-Qwen and compared it with several other models. LLaVA-Qwen uses the more powerful text-image matching encoder, SigLIP, which may synergize with our model to bring further improvements. Due to time constraints, we trained on our small datasets (300k LLaVA-1.5 image data + LLaVA-Hound-300k video data, totaling 600k), sampling 64 frames per video. The results are as follows (all compared models are based on LLaVA-Qwen2):
>
> | Method   | SFT Data | Context | VCGBench-avg |ActivityNet-QA |VideoMME-overall|VideoMME-Long|
> |:--------|:------:|:------:|:------:|:------:|:------:|:------:|
> | LLaVA-Qwen2   | 760K   | 8k    | 3.32 | 51.3 |49.5 | 43.4|
> | LongVA   | 760K   | 224k    | 3.58 | - |52.6 | 46.2|
> | LLaVA-OneVision   | 4.8M   | 8k    | 3.51 | 56.6 |**58.2** | 46.7|
> | PPLLaVA   | 600k   | 2k    | **3.63** |**57.0** |  55.2 | **47.1**|
>
> It can be observed that PPLLaVA remains highly competitive on the LLaVA-Qwen baseline. Despite having much less training data and shorter context length than LLaVA-One-Vision, it still achieves comparable performance. To be honest, we acknowledge that the changes from LLaVA-Qwen to LLaVA-Next are quite significant. Not only has the visual encoder changed from CLIP to SigLIP, but the LLM has also shifted from Vicuna to the much stronger Qwen2. Therefore, it is difficult for us to quantitatively determine how much of the improved results come from the stronger text-image matching encoder. However, from another perspective, on the LLaVA-Next baseline, PPLLaVA shows relative improvements of 0.23 and 2.6 on VCGBench and VideoMME, respectively. On LLaVA-Qwen, the relative improvements are 0.31 and 5.7. The relative improvements exclude the impact of the LLM in some degree, indicating that the combination of a stronger text-image matching encoder and PPLLaVA indeed further improves the model's performance.

---

> > ### Comment · Reviewer_1Y1n · 2024-11-27
> >
> > Thank the authors for the response. After reading the reply, I believe the paper benefits the community. I will keep my score.

---

> > > ### Author Response · Authors · 2024-12-02
> > > **Response to Reviewer 1Y1n**
> > >
> > > Thank you for your positive feedback. We greatly appreciate your continued support and the time you’ve dedicated to reviewing our work.

---

> ### Author Response · Authors · 2024-11-23
> **Looking forward to a reponse before the deadline**
>
> Dear Reviewer 1Y1n,
>
> We want to send you a friendly reminder that the stage of discussion will be completed soon.
>
> It is greatly appreciated if you are willing to reconsider your score based on our responses, and we really want to know whether our responses address your concerns. If there is any other concern that we could not address in the response, please feel free to let us know and we would be happy to provide further explanation.
>
> With sincere regards,
>
> Authors of Paper 1865

---

### Author Response · Authors · 2024-12-02
**Final Author Response**

We appreciate all the responsible reviewers and chairs, as well as the time they’ve dedicated to reviewing our work.

It is a pity that Reviewer hDF1 and Reviewer Qkbp did not engage in any effective communication with us throughout the review process. To clarify any potential misunderstandings for others, we will provide a final response to the comments from the two reviewers.

Reviewer hDF1's review is the most absurd we have ever received. The majority of their concerns have already been emphasized multiple times in the paper, and the remaining issues show a lack of basic knowledge about video understanding. It seems that Reviewer hardly read our paper and lacks common sense in the field. Unsurprisingly,  Reviewer hDF1 also did not provide any feedback within 18 days.

Reviewer Qkbp's reason for rejecting our paper can be summarized in two points: 1) The performance on MSRVTT and MSVD is not better than SlowFast-LLaVA; 2) The use of DPO is not novel. For the first point, we have already responded that this is due to the unfair comparison regarding the GPT version. Additionally, we found that training-free methods (FreeVA, IGVLM, SlowFast LLaVA) generally perform well on these two datasets, often comparable to trained methods. Furthermore, recent well-known MLLMs such as LLaVA-Video, LLaVA-OneVision, and LongVU did not report results on these two datasets. Therefore, we believe that rejecting our paper based on results from two non-specialized benchmarks, especially under such unfair comparison conditions, is unjust. For the second point, we have never claimed that the use of DPO is the main contribution or innovation of our work. We are puzzled as to why more implementations and experiments are ultimately criticized for lacking novelty.

We had hoped that our rebuttal would clear up Reviewer Qkbp's misunderstandings. However, the response, "Thanks for your reply, I'll maintain my score," which came after 14 days, was the shortest response we have encountered. And then, there was no further communication.

We have remained patient and polite with these two reviewers. However, due to their passive behavior, we were unable to resolve any misunderstandings. Hence, it is greatly appreciated if our work can be fairly judged.

Finally, thank you again for your time and for being part of the review process.

With sincere regards,

Authors of Paper 1865

---

### Meta-Review · Area_Chair_4tv2 · 2024-12-20

**Metareview:**

The paper proposes PPLLaVA, a video LLM utilizing prompt-guided pooling to efficiently process short and long videos. The method demonstrates strong performance across multiple video benchmarks. However, reviewers raised several concerns, including incremental  novelty, CLIP dependency, fairness of comparisons and writting issues. Some of these concerns are addressed and confirmed by reviewers, while some of them are not well addressed.

While the contributions are promising, the paper would benefit from addressing these concerns, including a deeper exploration of motivation and novelty, and a more thorough comparison with related works and baselines. Given the average score of 4.80 and intense competition at ICLR, the AC recommends revising the paper and considering submission to a future conference.

**Additional Comments On Reviewer Discussion:**

Reviewers raised several concerns, including incremental  novelty, CLIP dependency, fairness of comparisons and writting issues. Some of these concerns are addressed and confirmed by reviewers, while some of them are not well addressed.

---

### Decision · Program_Chairs · 2025-01-22

Reject